# Semantic Temporal Abstraction via Vision-Language Model Guidance for Efficient Reinforcement Learning

**Tian-Shuo Liu**[1,2]\***Xu-Hui Liu**[1]\***Ruifeng Chen**[1,2]\***Lixuan Jin**[1]**, Pengyuan Wang**[1,2]**,**
**Zhilong Zhang**[1,2]**, Yang Yu**[1,2]†
[1]National Key Laboratory for Novel Software Technology, Nanjing University, China
&School of Artificial Intelligence, Nanjing University, China
[2]Polixir Technologies
{liuts, liuxh, chenrf}@lamda.nju.edu.cn, jinlx@smail.nju.edu.cn,
{wangpy, zhangzl}@lamda.nju.edu.cn, yuy@nju.edu.cn

## Abstract

Extracting temporally extended skills can significantly improve the efficiency of reinforcement learning (RL) by breaking down complex decision-making problems with sparse rewards into simpler subtasks and enabling more effective credit assignment. However, existing abstraction methods either discover skills in an unsupervised manner, which often lacks semantic information and leads to erroneous or scattered skill extraction results, or require substantial human intervention. In this work, we propose to leverage the extensive knowledge in pretrained Vision-Language Models (VLMs) to progressively guide the latent space after vector quantization to be more semantically meaningful through relabeling each skill. This approach, termed **V**ision-l**an**guage model guided **T**emporal **A**bstraction (**VanTA**), facilitates the discovery of more interpretable and task-relevant temporal segmentations from offline data without the need for extensive manual intervention or heuristics. By leveraging the rich information in VLMs, our method can significantly outperform existing offline RL approaches that depend only on limited training data. From a theory perspective, we demonstrate that stronger internal sequential correlations within each sub-task, induced by VanTA, effectively reduces suboptimality in policy learning. We validate the effectiveness of our approach through extensive experiments on diverse environments, including Franka Kitchen, Minigrid, and Crafter. These experiments show that our method surpasses existing approaches in long-horizon offline reinforcement learning scenarios with both proprioceptive and visual observations.

## 1 Introduction

Reinforcement learning (RL) (Sutton & Barto, 2018) has demonstrated remarkable success across a variety of domains, including robotics (Peng et al., 2020; Haarnoja et al., 2018), games (Silver et al., 2016; Zha et al., 2021), and combinatorial optimization (Geng et al., 2024; Wang et al., 2024). However, in complex environments with sparse rewards, learning efficiently over long horizons remains challenging. Temporal abstraction (Sutton et al., 1999; Villecroze et al., 2022; Park et al., 2023; Fu et al., 2024) provides a solution by breaking down complex, long-term problems into simpler subtasks, allowing agents to learn hierarchically, where credit assignment can be more easily managed at the segment level. These methods seek to extract temporally extended primitive structures from tasks into skills for further learning, while in real-world scenarios with logged data, this falls under the scope of offline RL (Ajay et al., 2021; Venkatraman et al., 2024). The agent learns to behave within the constraints of the skill context, which reduces the action search space.

---

*Equal contribution.
†Corresponding author.

Despite previous efforts in hierarchical or goal-conditioned RL focusing on skill extraction for downstream policy learning, creating useful and effective abstractions remains challenging. Some methods apply unsupervised methods to discover skills (Ajay et al., 2021; Venkatraman et al., 2024), which often result in fragmented segments. While heuristic approaches, such as using fixed horizons to define subgoals (Emmons et al., 2022; Ghosh et al., 2021; Yang et al., 2022), can lead to unmeaningful divisions. In long-horizon tasks, particularly in offline RL, misleading extraction of primitive options can result in poor value function learning and suboptimal policy learning. To address this issue, another line of research incorporates human annotations or language descriptions into skill extraction (Lee et al., 2019; Fu et al., 2024; Peng et al., 2024), but these methods are heavily reliant on human supervision, which contradicts the pursuit of fully automated intelligence.

Humans inherently excel at efficient learning by instinctively decomposing complex tasks into smaller, manageable components. Drawing inspiration from this phenomenon, we aim to extract interpretable segments that emulate this task decomposition process, focusing on identifying meaningful contexts that denote subtasks. These contexts serve as temporally extended signals, representing the current subtask or the next subgoal, and condition the learning of skills (low-level contextual policies) for effective task execution. *How can we discover more reasonable context and build skills without relying on cumbersome human labor?* One promising approach to addressing this challenge is the integration of reinforcement learning agents with the prior knowledge and reasoning capabilities of pretrained foundation models, such as Vision-Language Models (VLMs). These models, trained on large-scale Internet data, offer rich interpretations of the environment and semantic understanding, allowing agents to reason in a more meaningful and effective manner (Du et al., 2024; Mees et al., 2023).

Inspired by the success of VLM, we propose a novel approach that leverages the prior knowledge of VLMs to decompose trajectories into reasonable subsequences, termed **V**ision-**Lan**guage Model Guided **T**emporal **A**bstraction (**VanTA**). Our method progressively shapes the latent space after vector quantization to be more semantically meaningful by relabeling each segment. This process facilitates the discovery of more interpretable, task-relevant temporal segmentations, all while reducing the reliance on extensive manual intervention. Specifically, the latent space undergoes alternating updates in two stages: first, it commits to the vector-quantized results; then, after VLM assigns indices to the segments, the latent space is further refined. We employ a two-level policy extraction framework, where the high-level policy selects from a compact, discrete primitive skill space, and Implicit Q-Learning (IQL) (Kostrikov et al., 2022) is used for low-level policy training. From a theoretical standpoint, we demonstrate that our method, in a hierarchical manner, compresses the raw MDP action space and reduces the horizon. Compared to the primitive skill space without VLM guidance, VanTA reduces the complexity of low-level policy space, allowing the agent to address subtasks more efficiently in expectation. We empirically validate our approach through extensive experiments on environments including Franka Kitchen in D4RL (Fu et al., 2020), Minigrid (Chevalier-Boisvert et al., 2023), and Crafter (Hafner, 2021). These experiments show that our method outperforms previous approaches in long-horizon offline reinforcement learning scenarios for both proprioceptive and visual observations.

## 2 RELATED WORK

### 2.1 TEMPORAL ABSTRACTION IN HIERARCHICAL REINFORCEMENT LEARNING

Temporal abstraction is a key topic in hierarchical reinforcement learning. Many previous methods rely on unsupervised objectives to discover skills (Pertsch et al., 2020; Eysenbach et al., 2019; Singh et al., 2021), which can lead to scattered and unmeaningful skill extraction. To achieve more coherent skill discovery, some approaches incorporate human intervention (Lin et al., 2024; Oh et al., 2017; Xu et al., 2018). Then these skills are used for planning (Sharma et al., 2020), few-shot imitation learning (Nam et al., 2022), or online RL (Nachum et al., 2019). In contrast, our work leverages VLM-guided skill extraction to enhance the limited available data, reducing the need for extensive human intervention. While temporally extended and recurring structures have been effective in solving complex tasks in online RL, integrating them into offline RL is also beneficial (Dietterich, 1998; Sutton et al., 1999; Kulkarni et al., 2016). The main challenge lies in how to hierarchically decompose trajectories (Ajay et al., 2021; Jiang et al., 2023; Pertsch et al., 2020). Our method focuses on extracting more meaningful skill context from offline data via VLM guidance.

## 2.2 Knowledge Distillation from VLM for Reinforcement Learning

Vision-language models (VLMs) possess extensive world knowledge. The efficiency of reinforcement learning is crucial (Chen et al., 2024; Wang et al., 2022), and leveraging knowledge in VLMs can enhance it. VLMs are used to decompose high-level long-sequence tasks into multiple low-level executable step-by-step plans (Huang et al., 2022; Wang et al., 2023; Brohan et al., 2023). These plans then serve as conditions for downstream reinforcement learning tasks to guide RL learning (Huang et al., 2023). However, due to the limited reasoning abilities of VLMs, they struggle to accurately recognize dynamic environments and reason precisely. On the other hand, VLMs are also used to embed useful information such as instructions (Liu et al., 2022; Mees et al., 2023; Myers et al., 2023), feedback (Bucker et al., 2023) and data for world modeling (Lin et al., 2023). These approaches leverage (V)LMs to encode the semantic information of input text and images. Compared to traditional knowledge distillation methods, our approach uses VLMs as a discriminative model to assist in segmentation, which is more reliable than relying on their generation abilities (Pang et al., 2024).

## 3 Preliminaries

Let $M = (\mathcal{S}, \mathcal{A}, P, r, \gamma, \rho_0)$ represent a Markov Decision Process (MDP), where $\mathcal{S}$ is the state space, $\mathcal{A}$ is the action space, $P : \mathcal{S} \times \mathcal{A} \to \Delta(\mathcal{S})$ is the transition function ($\Delta(\cdot)$ is the probability simplex), $r : \mathcal{S} \times \mathcal{A} \to [0, R_{\max}]$ is the reward function, $\gamma \in [0, 1)$ is the discount faction and $\rho_0$ is the initial distribution over states. A policy $\pi : \mathcal{S} \to \Delta(\mathcal{A})$ describes a distribution over actions for each state. The goal of RL is to learn the best policy $\pi^*$ that maximizes cumulative discounted reward, i.e., $\sum_t \mathbb{E}_{a_t \sim \pi^*} \gamma^t r(s_t, a_t)$. The value function and Q function of policy $\pi$ are $V^\pi(s) = \sum_t \mathbb{E}_{a_t \sim \pi(s_t)}[\gamma^t r(s_t, a_t)|s_0 = s]$, $Q^\pi(s, a) = \sum_t \mathbb{E}_{a_t \sim \pi(s_t)}[\gamma^t r(s_t, a_t)|s_0 = s, a_0 = a]$. $V^*$ and $Q^*$ be the shorthand for $V^{\pi^*}$ and $Q^{\pi^*}$ respectively. We assume access to experience dataset $\mathcal{D} := \{\tau_i := (s_t, a_t, r_t, s_{t+1})_{t=0}^H\}_{i=1}^N$.

Vector Quantized Variational Autoencoder (VQ-VAE) (van den Oord et al., 2017) is a neural network architecture designed for unsupervised learning of latent representations. In VQ-VAE, the input data $x$ is mapped to a continuous latent space by the encoder $q_\psi$, which is then quantized to the nearest codeword $z$ in the discrete codebook embedding space $\mathcal{Z}$. This codeword, $c_j$, is indexed in the codebook $\mathcal{C}$, where the size of the codebook set is denoted as $|\mathcal{C}|$. The decoder $p_\phi$ maps the discrete code back to the output space, generating new samples. VQ-VAE updates the encoder, decoder, and codebook parameters with the loss function Eq. (1), respectively reconstruction loss, codebook loss and commitment loss:

$$\mathcal{L} = \hat{\mathbb{E}}_{x \sim \mathcal{D}} \left[ \|x - \hat{x}\|_2^2 + \|\mathbf{sg}(h) - z\|_2^2 + \beta \|h - \mathbf{sg}(x)\|_2^2 \right], \tag{1}$$

where $\hat{x} \sim p_\phi(\cdot|z), h \sim q_\psi(\cdot|x)$, and $\mathbf{sg}(\cdot)$ represents the stop-gradient operation. Additionally, we follow the technique from Gumbsch et al. (2024), which models sequential transitions in latent space by introducing $L_1$-regularization to context changes $\Delta h_t = \|h_{t+1} - h_t\|_1$, as the context code is intended to change relatively sparsely.

## 4 Method

In this section, we introduce VanTA, a novel framework for extracting skills from offline trajectory datasets $\mathcal{D}$, grounded in pretrained vision-language models (VLMs). We incorporate VLM into the skill context codebook $\mathcal{Z}$ update process, which distills states into a discrete space $\pi_\theta(z|s)$. Then conditioned on the extracted skill, low-level policy is further learned by offline RL.

Our framework, as illustrated in Fig. 1, consists of two stages. In the first stage, state sequences are initially segmented using vector quantization (VQ) techniques to generate the initial set of skills. These skills are then passed to the VLM, which relabels them based on its internal knowledge. The relabeled skills are subsequently used to update the VQ codebook following the method described in van den Oord et al. (2017). This process is repeated iteratively until convergence. In the second stage, the low-level policy is refined using the skills derived from the first stage. The following sections provide detailed insights into the implementation of our approach.

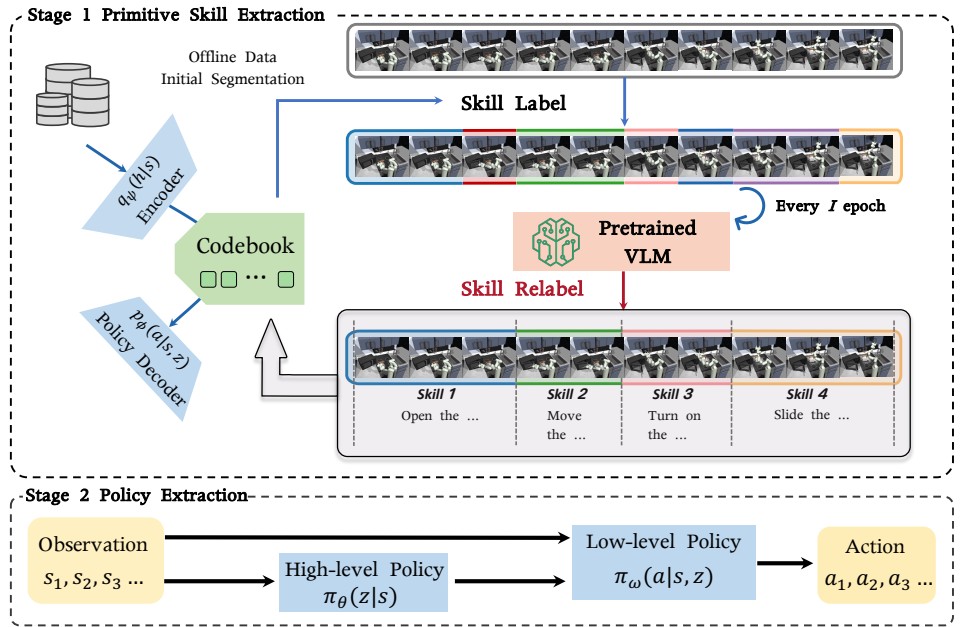

Figure 1: **V**ision-l**an**guage model guided **T**emporal **A**bstraction (VanTA) Overview: First stage for primitive skill extraction, second stage for policy extraction, conditioned on the extracted skill.

### 4.1 SKILL CONTEXT LEARNING

To better distill the skill context, we leverage the knowledge in the foundation model VLM. VQ provides a standard codebook $\mathcal{Z}$, but directly querying each image may lead to hallucinations due to the lack of clear meaning in a single observation. Additionally, this approach is inefficient when handling large offline datasets. Therefore, we start with the VQ method to initialize trajectory segments. The input for VQ is a single observation. By projecting consecutive observations into a single skill context, segments are formed, which serve as a proper initialization for querying the VLM. We propose a modification to the original VQ-VAE architecture, to learn text-aligned temporal abstractions of trajectories for decision-making policies. The original decoder is replaced with a policy decoder $p_\phi(a|s, z)$, as temporal abstractions focus on behavior information for policy learning rather than sharing the reconstruction objective.

As the encoder $q_\psi(h|s)$ embeds the state into the continuous latent space, it is then projected into a discrete code vector $z_j$, where the index $j$ corresponds to codebook $c_j$ in $\mathcal{C}$. Consecutive states assigned to the same code index naturally segment the trajectory into sub-trajectories, where the initial and terminal states of each sub-trajectory define the boundaries of a primitive skill. To ground the codebook embedding space with semantic meaning, we query the VLM as follows: $j = \text{VLM}(\bar{s})$, where $\bar{s}$ represents the initial and terminal states of the primitive skill. The returned $j$ is the identified index. We adopt the update rule from van den Oord et al. (2017); Roy et al. (2018); Razavi et al. (2019) to update the $c_j$ embedding $z_j$, applying an exponential moving average with a smaller decay rate after querying the VLM:

$$z_j \leftarrow z_j \cdot \lambda + \sum_{n_j} h_j(1 - \lambda), \tag{2}$$

where $\sum_{n_j}$ represents all projected embeddings assigned to index $j$ by the VLM. We denote the *EMA* target of $(\sum_{n_1} h_1(1 - \lambda), ..., \sum_{n_j} h_j(1 - \lambda), ...)$ as $e$, where $j$ ranges from 1 to $|\mathcal{C}|$ in Eq. (3).

Scattered or unmeaningful skills are common during unsupervised learning. Leveraging knowledge in the VLM, primitive skills can be merged, and misleading indices corrected. This alternation process gradually makes the segmentation more semantically meaningful. Exception handling for VLM is crucial, as the output of VLM can be stochastic, often providing misleading answers when

an image only partially depicts the subtask. We thereby add one option for VLM to choose as no suitable skill. During optimization, we increase the smoothness loss to improve consistency with surrounding states, incorporating scattered observations into neighboring skills in subsequent iterations. The final loss for training encoder $q_\psi$, decoder $p_\phi$ is defined as:

$$\mathcal{L}_{\psi,\phi} = \hat{\mathbb{E}}_{s,a \sim \mathcal{D}} \left[ \|a - \hat{a}\|_2^2 + \beta \|h - \mathbf{sg}(z)\|_2^2 + \|z - \mathbf{sg}(h)\|_2^2 + \|z - e\|_2^2 + \gamma \|\Delta h\|_1 \right], \quad (3)$$

where $\mathbf{sg}(\cdot)$ represents the stop-gradient operation, and $\Delta h$ denotes the smoothness loss, which ensures representation consistency between consecutive states, following the approach in Gumbsch et al. (2024). And $z \in \mathcal{Z}$ is the latent variable discretized by vector quantization in this loss function.

## 4.2 LOW-LEVEL POLICY EXTRACTION

We collect a replay buffer $\mathcal{B}$ from the dataset $\mathcal{D}$, consisting of segmented sub-trajectories, and store transition tuples $\{\sigma_i := (s_t, a_t, z_t, r_{t:t+K-1}, s_{t+K})_{t=0}^{C_i}\}_{i=0}^M$ from $\sigma_i \sim \mathcal{B}$. Here, $s_t$ and $a_t$ represent the state-action pairs from $\mathcal{D}$, $z_t$ is the latent variable labeled by our primitive skill extraction method, $r_{t+K-1}$ denotes the $\gamma$-discounted sum of rewards accumulated over $K-1$ time steps starting from time step $t$ in $\sigma_i$, and $s_{t+K}$ represents the initial state of the next skill. We first update the high-level Q-function, which is conservatively learned using temporal-difference error at the skill level. A batch of primitive transitions is sampled from $\mathcal{B}$ for maximizing the cumulative return of the primitive skills. Since the latent space in the codebook is quantized, we apply discrete Deep Q-learning under offline setting in a conservative manner (Kumar et al., 2020). The Bellman error for the Q-update in high-level policy learning can be expressed as:

$$Q(s_t, z_t) \leftarrow r_{t:t+K-1} + \gamma^K Q(s_{t+K}, \arg\max_{z \sim \pi_\theta(z|s_{t+K})} Q(s_{t+K}, z)), \quad (4)$$

Once the high-level policy is applied to generate candidate primitive skills during evaluation, the final policy for VanTA generates low-level actions conditioned on both the primitive skill and the observation. The low-level policy is learned using a behavior cloning-style algorithm, where the policy network approximates the action logits from the offline dataset:

$$\min_\omega J(\omega; \mathcal{D}) = \hat{\mathbb{E}}_{\tau \sim \mathcal{D}} \left[ -\sum_{t=0}^{H-1} \log \pi_\omega(a_t|s_t, z_t) \right], \quad (5)$$

where $z$ is the latent space label assigned to the state. Additionally, we apply importance weighting during the update, using $\exp(Q(s,a) - V(s))$ (Kostrikov et al., 2022) with a certain coefficient.

## 5 ANALYSIS

Intuitively, more semantically meaningful skills can lead to better policies. To theoretically validate this intuition, we analyze the relationship between algorithm performance and the learned skills. We compare VanTA with reinforcement learning in the original space or hierarchical reinforcement learning methods that lack external semantic information guidance in this section. Our algorithm is divided into two parts: offline reinforcement learning for the high-level policy and weighted behavior cloning for the low-level policy. First, we focus on the offline reinforcement learning part. In the offline setting, the performance of the policy is highly related to the dataset. Therefore, we introduce an assumption on the quality of the dataset as previous works did (Chen & Jiang, 2019; Liu et al., 2024). We also give the definition of *admissible distributions*.

**Definition 5.1** (Admissible distributions). We say a distribution $\nu \in \Delta(\mathcal{S} \times \mathcal{A})$ is admissible in MDP $M$, if there exists $t \geq 0$ and a policy $\pi$ such that $\nu(s, a) = \Pr[s_t = s, a_t = a|s_0 \sim \rho_0, \pi]$.

In other words, admissible distributions are distributions that can be generated by some policy under the given MDP.

**Assumption 5.2** (Concentratability coefficient). Given the dataset $D$, let the distribution of the dataset be $\mu(s, a)$. We assume that there exists $C < \infty$ s.t. for any admissible $\nu$, $\frac{\nu(s,a)}{\mu(s,a)} \leq C$, $\forall (s, a) \in \mathcal{S} \times \mathcal{A}$.

The assumption means the distribution of the offline dataset should be able to include all the state-action pairs. Chen & Jiang (2019) gives the sample complexity of performing fitted Q iteration under this assumption.

**Theorem 5.3** (Theorem 2 of Chen & Jiang (2019)). *Given a dataset $D$ with sample size $|D| = n$ and $\mathcal{F}$ satisfies completeness, i.e., $\forall f \in \mathcal{F}, \mathcal{T}f \in \mathcal{F}$, w.p. $\geq 1 - \delta$, the output policy of Fitted Q Iteration after $k$ iterations, $\pi_k$, satisfies $v^* - v^{\pi_k} \leq \epsilon V_{\max}$ when $k \to \infty$ and $n = O\left(\frac{C \log \frac{|\mathcal{F}|}{\delta}}{\epsilon^2 (1-\gamma)^4}\right)$.*

We assume the size of the function class $|\mathcal{F}|$ is influenced by the dimensions of the state space and action space, i.e., $|\mathcal{F}| = g(|\mathcal{S}|, |\mathcal{A}|)$. This assumption is reasonable because the complexity of function approximation is closely tied to the number of possible state-action pairs. Specifically, in a tabular setting, the size of the function class can be represented as: $|\mathcal{F}| = (|\mathcal{S}||\mathcal{A}|)^R$, where $|\mathcal{S}|$ is the cardinality of the state space, $|\mathcal{A}|$ is the cardinality of the action space, and $R$ represents the range of possible values that the Q-function can take. If we do Q iteration without establishing the hierarchical structure, the sample complexity of the learning process is indicated by Thm. 5.3.

In our setting, the high-level policy outputs not the actions but the skills, the low-level policy learns with weighted BC conditioned on the skill output by the high-level policy. To facilitate our analysis, we assume the length of skills is the same, denoted as $K$. The learning process of the high-level policy is similar to fitted Q iteration except the action space is replaced with the skill space and the horizon changes correspondingly. Therefore, we get the sample complexity of high-level policy directly from Thm. 5.3.

**Corollary 5.4.** *Given a dataset $D$ with sample size $|D| = n$ and $\mathcal{F}$ satisfies completeness, w.p. $\geq 1 - \delta$, the output policy of FQI after $k$ iterations, $\pi_k$, satisfies $\hat{v}^* - v^{\pi_k} \leq \epsilon V_{\max}$ when $k \to \infty$ and $n = O\left(\frac{C \log \frac{g(|\mathcal{S}|,|\mathcal{Z}|)}{\delta}}{\epsilon^2 (1-\gamma^K)^4}\right)$, where $\hat{v}^*$ is the optimal policy conditioned on the low-level policy.*

Note that $v^*$ in Thm. 5.3 is replaced with $\hat{v}^*$, this is because the quality of the skills is also an influence factor of the performance. Then we need to determine the difference between $\hat{v}^*$ and $v^*$.

Note that the low-level policy is trained by weighted behavior cloning, the optimal policy of weighted behavior cloning is Kostrikov et al. (2022): $\tilde{\pi}^*(a|s) \propto \mu(a|s) \exp\left(\beta(\hat{Q}(s,a) - \hat{V}(s))\right)$. We denote the value induced by the optimal policy as $\tilde{v}^*$. Therefore, $v^* - \hat{v}^*$ can be decomposed as $v^* - \tilde{v}^* + \tilde{v}^* - \hat{v}^*$. Let $\Delta(\beta, \hat{Q}, \hat{V})$ denote $v^* - \tilde{v}^*$, this is because this term is only related to $\beta$, $\hat{Q}$ and $\hat{V}$. If $\hat{Q} = Q^*$, $\hat{V} = V^*$, and $\beta \to \infty$, then $\tilde{v}^* \to v^*$. The last term we need to determine is $\tilde{v}^* - \hat{v}^*$. We give the result here, and the proof is deferred to Appx. A.

**Lemma 5.5.** *For any $\tilde{\pi}^* \in \Pi$, weighted BC algorithm ensures that with probability at least $1 - \delta$,*

$$\tilde{v}^* - \hat{v} \leq O(1) \cdot \sqrt{1 + \hat{Q}_{Var}(\beta)} \frac{H}{K} \sqrt{\frac{\sigma_K^2 \log(|\Pi|\delta^{-1})}{n}} + O(R \log(n)) \cdot (1 + \hat{Q}_{Var}(\beta)) \frac{H}{K} \frac{\log(|\Pi|\delta^{-1})}{n},$$

*where $\hat{v}$ is the value function induced by the learnt policy $\hat{\pi}$, $n$ is the number of samples in the dataset, $\hat{Q}_{Var} = Var\left(\frac{\exp(\beta(\hat{Q}(s,a) - \hat{V}(s)))}{Z(s)}\right)$, $Z(s)$ is the normalization factor, and*

$$\sigma_K^2 = \sum_{h=1}^{K} \mathbb{E}^{\pi^*}\left[(Q_t^{\pi^*}(s_t, \pi^*(s_t)) - Q_t^{\pi^*}(s_t, a_t))^2\right].$$

Note that a semantic skill implies temporal correlations of action sequences, which means the policy space of a semantic skill is not the same as the original policy space. For ease of discussion, we can make the following assumptions:

1. **Original policy space (no temporal correlation assumptions):** In the original policy space, there is no temporal correlation between action sequences. The agent can freely choose each action without considering the relationships between actions over time. This space is denoted as $\Pi_{\text{ori}}$.

2. **Semantic skill policy space (with temporal correlation):** Introducing temporal correlation means that actions at the current time step are influenced not only by the current state but also by previous actions. This will reduce the size of the strategy space, as not all action sequences are valid.

Suppose the autocorrelation coefficient of action sequences generated by the policy belongs to the new policy space is larger than $\alpha$, then the new space can be denoted as $\Pi_\alpha = \{\pi \in \Pi_{\text{ori}} | \rho(\pi) > \alpha\}$, where $\rho(\pi)$ is the autocorrelation coefficient of the action sequence generated by policy $\pi$.

Combine the results together, we have

**Theorem 5.6.** *Given a dataset $D$ with sample size $n$, under the hierarchical learning process, with probability as least $1 - \delta$,*

$$v^* - v^{\pi_k} \leq O\left(\sqrt{\frac{C\log\frac{|g(|\mathcal{S}|,|\mathcal{Z}|)|}{\delta}}{(1-\gamma^K)^4 n}}\right) \cdot V_{\max} + O(1) \cdot \sqrt{1 + \hat{Q}_{Var}(\beta)} \frac{H}{K} \sqrt{\frac{\sigma_K^2 \log(|\Pi_\alpha|\delta^{-1})}{n}}$$

$$+ O(R\log(n)) \cdot (1 + \hat{Q}_{Var}(\beta)) \frac{H}{K} \frac{\log(|\Pi_\alpha|\delta^{-1})}{n} + \Delta(\beta, \hat{Q}, \hat{V}).$$

Compare Thm. 5.6 with Thm. 5.3, the hierarchical structure reduces the size of Q function class from $g(|\mathcal{S}|, |\mathcal{A}|)$ to $g(|\mathcal{S}|, |\mathcal{Z}|)$, and the scale induced by horizon is reduced from $(1 - \gamma)^4$ to $(1 - \gamma^K)^4$. However, it also introduces three other terms. Generally, $\Delta(\beta, \hat{Q}, \hat{V})$ is not large, as the weighted BC algorithm has already achieved good performance (Nair et al., 2020; Zhao et al., 2022; Kostrikov et al., 2022) and $\Delta(\beta, \hat{Q}, \hat{V})$ is even smaller because there is no BC loss and the high-level policy is assumed as optimal. If the hierarchical structure of the problem is good, for example, the action sequence of the skill has high temporal correlation and the middle two terms are also small, then the performance bound of hierarchical methods is better than the original one.

Furthermore, Thm. 5.6 provides us with an approach to determine the performance of different hierarchical methods. If the action sequence of the learned skill has high temporal correlation, $|\Pi_\alpha|$ can be much smaller, then the performance bound is much better. Intuitively, the skills with semantic information are more temporal correlated, which explains why our method can be better than previous methods. We test the temporal correlation in the experiment Sec. 6.5.

Table 1: Performance on Franka Kitchen, MiniGrid and Crafter environments, averaged over 5 random seeds. The best results are **bolded** and the second-best results are underlined.

| Task Name | BC | CQL | IQL | RvS | GCSL | WGCSL | LDCQ | VanTA (Ours) |
|---|---|---|---|---|---|---|---|---|
| kitchen-complete-v0 | 65.0 | 43.8 | 62.5 | 50.2 | 58.6 | 57.7 | 52.8 | **69.2±8.5** |
| kitchen-partial-v0 | 38.0 | 50.1 | 46.3 | 60.3 | 55.0 | 59.4 | 67.8 | **71.2±5.7** |
| kitchen-mixed-v0 | 51.5 | 52.4 | 51.0 | 51.4 | 56.2 | 49.6 | 62.3 | **68.5±4.4** |
| MiniGrid-DoorKey-6x6-complete-v0 | **0.92** | 0.67 | 0.91 | 0.89 | 0.82 | 0.85 | 0.86 | **0.92±0.03** |
| MiniGrid-DoorKey-6x6-mixed-v0 | 0.70 | 0.61 | 0.72 | 0.67 | 0.59 | 0.79 | 0.72 | **0.80±0.17** |
| MiniGrid-DoorKey-8x8-complete-v0 | 0.88 | 0.43 | 0.89 | **0.94** | 0.87 | 0.76 | 0.70 | 0.92±0.07 |
| MiniGrid-DoorKey-8x8-mixed-v0 | 0.43 | 0.30 | 0.47 | 0.32 | 0.39 | 0.44 | 0.38 | **0.51±0.18** |
| MiniGrid-KeyCorridorS3R3-complete-v0 | 0.47 | 0.62 | 0.72 | 0.64 | 0.55 | 0.57 | 0.69 | **0.81±0.06** |
| MiniGrid-KeyCorridorS3R3-mixed-v0 | 0.09 | 0.23 | 0.51 | 0.42 | 0.21 | 0.23 | 0.38 | **0.61±0.12** |
| MiniGrid-RedBlueDoors-6x6-complete-v0 | 0.80 | 0.51 | **0.90** | 0.78 | 0.73 | 0.61 | 0.64 | 0.85±0.11 |
| MiniGrid-RedBlueDoors-6x6-mixed-v0 | 0.64 | 0.47 | 0.69 | 0.42 | 0.72 | 0.51 | 0.44 | **0.73±0.20** |
| Crafter-partial | 2.69 | 1.73 | 2.75 | 2.11 | —— | —— | 0.96 | **5.46** |

## 6 EXPERIMENTS

In this section, we empirically validate the effectiveness of VanTA in providing a hierarchical structure that improves agent learning performance. Sec. 6.1 evaluates the performance of offline RL methods against hierarchical goal-conditioned approaches on tasks with hierarchical sub-tasks. Sec. 6.2 explains VanTA's superior performance by showcasing how it segments trajectories into interpretable skills, outperforming traditional VQ methods. Sec. 6.3 reinforces this with an ablation study, comparing codebooks with and without VLM guidance, showing that VLM guidance enhances learning, while its absence hinders performance. Sec. 6.4 demonstrates VanTA's strength in data-scarce scenarios, leveraging pretrained VLM knowledge to augment datasets and improve results. Finally, Sec. 6.5 verifies our theory and demonstrate the higher temporal correlation of VanTA. For experimental benchmarks, we conduct our experiments on three long-horizon domains: Franka Kitchen (Gupta et al., 2019) for manipulation tasks, MiniGrid (Chevalier-Boisvert et al., 2023)

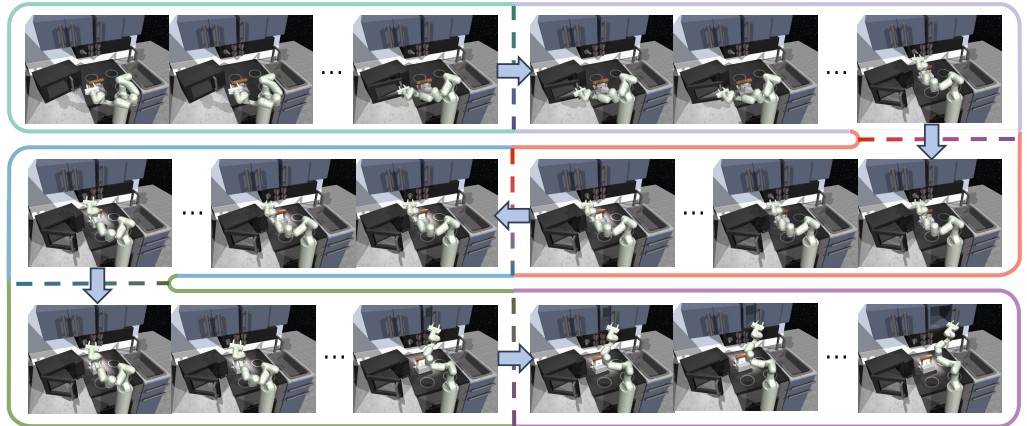

Figure 2: Skill extraction results without VLM guidance; no clear semantic meaning in each skill.

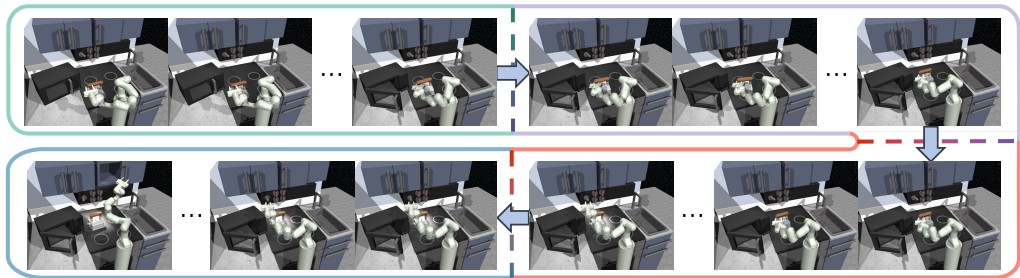

Figure 3: Skill extraction results using VanTA: sequentially **1) open the microwave, 2) move the kettle, 3) rotate the knob, and 4) open the cabinet**.

for navigation, and Crafter (Hafner, 2021), an open-world survival game. Further implementation details, including model architectures and hyperparameter settings, are provided in Appx. B.5.4. We first introduce the environment setup.

**Franka Kitchen**: This environment is initially introduced in Gupta et al. (2019). In our experiments, we use offline data from the Fu et al. (2020) benchmark, which includes three types of datasets collected from human demonstrations, with visual observations rendered using OpenAI's Gym framework rendering functions (Brockman et al., 2016) as input for VLMs.

**MiniGrid**: Chevalier-Boisvert et al. (2023) is a lightweight grid-world environment suite with OpenAI Gym interfaces. All tasks fall within the grid navigation domain, where the agent starts from an initial position, picks up an object, and navigates to the goal destination with sparse rewards.

**Crafter**: (Hafner, 2021) is a 2D open-world survival game benchmark with visual inputs that evaluates a wide range of general abilities within a single environment. We collect data using intermediate model training checkpoints from online reinforcement learning with PPO (Schulman et al., 2017).

### 6.1 EVALUATION ON BENCHMARKS

We empirically evaluate our two-level policy extraction method on diverse tasks, with all of them providing sparse reward feedback. Prior approaches include typical offline reinforcement learning or imitation learning methods including Behavior Cloning (BC), Conservative Q-Learning (CQL) (Kumar et al., 2020), and Implicit Q-Learning (IQL) (Kostrikov et al., 2022), along with goal-conditioned variants RvS (Emmons et al., 2022), GCSL (Ghosh et al., 2021), WGCSL (Yang et al., 2022) and LDCQ (Venkatraman et al., 2024), which use goal-setting methods like hindsight goal relabeling or prior-shaped latent space. As shown in Fig. 1, VanTA outperforms the baselines in most tasks, which demonstrates that the extracted primitive skills aid in policy learning. Since Crafter is an open-world, vision-based environment, manually setting a desirable observation as the

goal in GCSL and WGCSL is infeasible. Investigating the Kitchen tasks, when performing the final subtask of opening the cabinet, the agent selects actions from a higher-level, discrete and internally-compact skill space, where the robot arm has already been raised, rather than searching through the full original action space. In Crafter, which involves more subtasks, the extracted skills motivate the agent to explore a wider range of tasks, as demonstrated in Tab. 6. A similar conclusion can be drawn in the KeyCorridor environment, which is comparatively more complex in MiniGrid.

## 6.2 VISUALIZATION OF SEGMENTATION RESULT

In this section, we compare the segmentation results with the vanilla VQ method to demonstrate VanTA's superiority in interpretable skill extraction. In the kitchen-complete-v0 task, the agent sequentially opens the microwave door, moves the kettle, turns on the light, and finally opens the cabinet. The skill assignment results, shown in Fig. 3, display the first two observations and the terminal observation for each skill, highlighting how VanTA reduces the incoherence of primitive skills. In contrast, the vanilla VQ method may abruptly shift to a new skill in successive time steps, as seen during the microwave door opening in Fig. 2. Intuitively, although VQ occasionally provides imperfect initialization, VLM intervention quickly corrects these erroneous assignments and merges meaningful skills. Additional segmentation visualizations can be found in Appx. B.5.1.

## 6.3 ABLATION

**VanTA v.s. VanTA w/o VLM guidance** We next evaluate whether VLM guidance significantly improves learning in hierarchical models. To test this, we replace the indices assigned to primitive skills with those from the VanTA without VLM guidance method. As shown in Fig. 4, in the Franka Kitchen environment, VLM-guided skills are conducive to policy learning, demonstrating superiority across all environments. When compared with the performance of offline reinforcement learning baselines shown in Tab. 1, the performance of using non-VLM-guided extracted skill contexts to supplement state information for policy learning even shows a drop. This suggests that an erroneous or broken codebook introduces challenges in two-level policy learning, especially making it more difficult to learn primitive skills. Besides, using

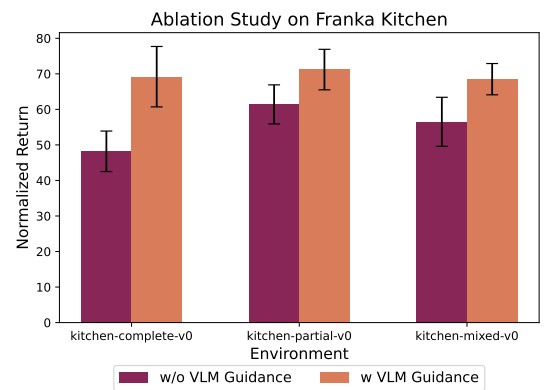

Figure 4: Comparison of normalized return between VanTA and VanTA without VLM guidance in the Franka Kitchen environment.

irrational primitive skills to augment the state also may not be beneficial. Additional ablation results are available in Appx.B.5.3.

## 6.4 EXPERIMENTS IN LOW-DATA REGIME

Leveraging the rich knowledge pretrained in VLMs enhances the efficiency of reinforcement learning, as it is a supplement to the limited available data. We further investigate whether VanTA enables more efficient learning with limited data using varying data ratios: 100%, 50%, and 10%. As shown in Tab. 2, VanTA maintains its performance advantage and degrades relatively more slowly as the available data decreases. Notably, on the kitchen-complete-v0 task with only 10% of the data, the smallest dataset in our experiment, VanTA achieves a 28.6% improvement, highlighting its ability to boost the efficiency of offline reinforcement learning.

## 6.5 REDUCTION IN COMPLEXITY

As discussed in Sec. 5, the performance of the algorithm is influenced by the size of the policy space it identifies. This size is largely determined by temporal correlations in the data. Therefore, in this section, we calculate the temporal correlation to assess the identified policy space. We assume that the original policy space is derived from the original dataset. To analyze this, we randomly sample

Table 2: Performance with reduced offline data usage in Kitchen, averaged over 5 random seeds.

| Task Name | BC | CQL | IQL | RvS | GCSL | WGCSL | LDCQ | VanTA (Ours) |
|---|---|---|---|---|---|---|---|---|
| kitchen-complete-v0 (100%) | 65.0 | 43.8 | 62.5 | 50.2 | 58.6 | 57.7 | 52.8 | **69.2±8.5** |
| kitchen-complete-v0 (50%) | 23.6 | 18.6 | 35.4 | 36.0 | 28.3 | 29.5 | 32.5 | **44.2±6.2** |
| kitchen-complete-v0 (10%) | 7.6 | 2.7 | 13.3 | 5.0 | 7.3 | 9.1 | 12.5 | **17.1±7.7** |
| kitchen-partial-v0 (100%) | 38.0 | 50.1 | 46.3 | 60.3 | 55.0 | 59.4 | 67.8 | **71.2±5.7** |
| kitchen-partial-v0 (50%) | 27.1 | 12.8 | 39.7 | 15.6 | 12.9 | 25.8 | 17.7 | **42.8±5.1** |
| kitchen-partial-v0 (10%) | 19.1 | 9.2 | 33.2 | 19.5 | 14.2 | 36.9 | 21.3 | **37.5±3.3** |
| kitchen-mixed-v0 (100%) | 51.5 | 52.4 | 51.0 | 51.4 | 56.2 | 49.6 | 62.3 | **68.5±4.4** |
| kitchen-mixed-v0 (50%) | 30.4 | 14.4 | 25.5 | 20.8 | 16.9 | 31.4 | 21.2 | **45.8±5.2** |
| kitchen-mixed-v0 (10%) | 21.2 | 8.7 | 23.4 | 20.0 | 15.2 | 27.8 | 11.4 | **29.9±3.0** |

fractions of the trajectories from the dataset and calculate their temporal correlation. This allows us to obtain the distribution of temporal correlations within the original policy space. For various hierarchical algorithms, including VanTA and VanTA without VLM guidance, we similarly calculate the temporal correlations of the identified skills. By comparing the range of temporal correlations from these algorithms with the original distribution, we can estimate the ratio between the size of the policy space identified by each algorithm and that of the original policy space.

To approximate the distribution of the temporal correlation of the original policy space, we sample 1000 fractions of trajectories and use Kernel Density Estimation (KDE). We also use 100 samples to approximate the range of temporal correlation of low-level policies. The temporal correlation is measured by the autocorrelation coefficient. For an action sequence $\{a_1, a_2, \ldots, a_K\}$, the autocorrelation coefficient is defined as: $r_k = \left( \sum_{t=k+1}^{K} (a_t - \bar{a})(a_{t-k} - \bar{a}) \right) / \sum_{t=1}^{K} (a_t - \bar{a})^2$. In the experiment, we set $k = 1$, with the results presented in Tab. 3. VanTA achieves a higher autocorrelation coefficient than the non-VLM-guided version across all three environments. On average, the policy space is reduced to 64% without VLM guidance and further to 54% with VLM guidance. This reduction in policy space complexity explains VanTA's superior performance and supports the theory discussed in Sec. 5.

Table 3: The comparison of the policy spaces of the low-level policy learned using VanTA without VLM guidance alone and those learned using VanTA.

| Task Name | VanTA w/o VLM | | VanTA | |
|---|---|---|---|---|
| | $r_1$ | Ratio | $r_1$ | Ratio |
| kitchen-complete-v0 | 0.42 | 60% | 0.69 | 41% |
| kitchen-partial-v0 | 0.44 | 80% | 0.59 | 69% |
| kitchen-mixed-v0 | 0.67 | 53% | 0.68 | 52% |

# 7 CONCLUSION

**Discussion.** We introduce VanTA, a method for extracting discrete, task-relevant and semantic skills from offline data with the guidance of pretrained Vision-Language Models (VLMs), which improves learning efficiency. This grounding of the codebook in semantic knowledge facilitates subsequent offline RL. In future work, we primarily aim to explore temporal abstraction with VLM for offline preference learning (Zhang et al., 2024) and cooperative multi-agent systems (Yuan et al., 2023). Our initialization technique has significantly reduced query complexity, which we aim to reduce further in future work. However, VanTA has certain limitation. In our experiments, proper initialization is essential for effective VLM guidance; without it, significant query complexity can arise, making it difficult to extract interpretable meaning from a single observation. Additionally, if mode collapse occurs, a common issue in unsupervised learning (Srivastava et al., 2017), VLM is unable to rectify these errors. Therefore, in future work, we aim to explore simpler yet more effective initialization methods.

## REPRODUCIBILITY STATEMENT

In this study, to ensure the reproducibility of our approach, we provide key information from our submission as follows.

1. **Training Algorithm.** We provide the architecture in Sec. 4 and pseudo-code of our approach in Appx. 1.
2. **Source Code.** We have submitted the source code of our approach in the supplementary materials. For BC, IQL (Kostrikov et al., 2022), CQL (Kumar et al., 2020) in Tab. 1, we apply the code from https://github.com/yihaosun1124/OfflineRL-Kit. For the other baselines in Sec. 6, we follow the official code and build upon them. We use the code from Emmons et al. (2022) for RvS. available at https://github.com/scottemmons/reinforcement-learning-via-supervised-learning. For GCSL (Ghosh et al., 2021) and WGCSL (Yang et al., 2022), we modify upon https://github.com/YangRui2015/AWGCSL. For LDCQ (Venkatraman et al., 2024), we build upon the official code, available at https://github.com/ldcq/ldcq.
3. **Experimental Details.** We list the detailed experiment settings, computational resources and hyperparameters in Appx. B
4. **Theoretical Proofs.** We provide the missing proofs in Appx. A.

## AUTHOR CONTRIBUTIONS

Y. Y.: led the direction, oversaw the project;

T.-S. L., X.-H. L., R. C.: designed the method;

X.-H. L.: proved the theory;

T.-S. L., P. W.: implemented the codes;

T.-S. L.: designed the experiments;

L. J.: conducted experiments;

T.-S. L.: analyzed the results;

T.-S. L., R. C.: wrote the paper;

P. W., Z. Z.: revised the paper;

T.-S. L.: discussed and wrote the response;

## ACKNOWLEDGEMENTS

This work was supported by the NSFC (62495093) and Jiangsu SF (BK20243039).

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

# A    MISSING PROOFS

We state the proof of Lemma 5.5 here. Foster et al. (2024) provided the performance bound of BC algorithms:

**Theorem A.1** (Corollary 3.1 of (Foster et al., 2024)). *For any $\pi^* \in \Pi$, BC algorithm ensures that with probability at least $1 - \delta$,*

$$v^* - \hat{v} \leq O(1) \cdot \sqrt{\frac{\sigma_K^2 \log(|\Pi|\delta^{-1})}{n}} + O(R\log(n)) \cdot \frac{\log(|\Pi|\delta^{-1})}{n},$$

*where $\hat{v}$ is the value function induced by the learnt policy $\hat{\pi}$, $n$ is the number of samples in the dataset, and*

$$\sigma_K^2 = \sum_{h=1}^{H} \mathbb{E}^{\pi^*} \left[ (Q_t^{\pi^*}(s_t, \pi^*(s_t)) - Q_t^{\pi^*}(s_t, a_t))^2 \right].$$

Note that weighted BC does not share the same performance bound as BC. Given the same dataset, weighted BC has a higher variance, which increases the bound in Theorem A.1. Intuitively, applying weighted BC in a given dataset can be seen as applying BC in another data distribution at the expense of excluding the data that does not conform to the new data distribution. Therefore, to apply Theorem A.1 to our setting, we introduce the concept of effective sample size, under which the estimator of the unweighted distribution has the same variance as that of the weighted estimator under the original sample size.

**Lemma A.2.** *Suppose the distribution before adding weight is $p$, and after adding weight is $q$, the effective sample size is*

$$n_{eff} \approx \frac{n}{1 + Var\left(\frac{q}{p}\right)},$$

*where $n$ is the size of the original dataset.*

*Proof.* 1. Consider an unbiased estimator $\hat{\mu}$ obtained through resampling:

$$\hat{\mu} = \frac{1}{N} \sum_{i=1}^{n} w_i X_i$$

Here, $X_i$ are the observed values, and $w_i$ are the weights reflecting the resampling. If all weights $w_i$ are equal (e.g., $w_i = 1$), this corresponds to simple random sampling. However, with weighted sampling, unequal weights can reduce the "effective" information in the sample.

2. To derive effective sample size, we introduce the variance of the estimator:

$$\text{Var}(\hat{\mu})_{\text{weighted}} = \frac{1}{n^2} \sum_{i=1}^{n} w_i^2 \text{Var}(X_i) = \frac{1}{n^2} \sum_{i=1}^{n} w_i^2 \sigma^2,$$

where $\sigma^2$ is the variance of a single sample $X_i$.

In the absence of weighting (i.e., $w_i = 1$), the variance becomes:

$$\text{Var}(\hat{\mu})_{\text{unweighted}} = \frac{\sigma^2}{n}.$$

3. Replace $n$ in $\text{Var}(\hat{\mu})_{\text{unweighted}}$ as $n_{\text{eff}}$, we make the two variances to be the same by tuning $n_{\text{eff}}$, i.e.,

$$\frac{1}{n^2} \sum_{i=1}^{n} w_i^2 \sigma^2 = \frac{\sigma^2}{n_{\text{eff}}}$$

$$n_{\text{eff}} = \frac{n^2}{\sum_{i=1}^{n} w_i^2}$$

In other words, effective sample size is the total sample size $n$ squared divided by the sum of the squared weights.

4. Given that $w_i = \frac{q(x_i)}{p(x_i)}$, we have:

$$\frac{1}{N} \sum_{i=1}^{N} w_i = 1 \quad \Rightarrow \quad \sum_{i=1}^{N} w_i = N$$

We need to evaluate the sum of the squared weights:

$$\sum_{i=1}^{N} w_i^2$$

This can be related to the variance of the weights. Recall that the variance of a set of values is defined as:

$$\text{Var}(w) = \frac{1}{N} \sum_{i=1}^{N} (w_i - \overline{w})^2$$

where $\overline{w}$ is the mean of the weights. Given that the weights are normalized to have a mean of 1 ($\overline{w} = 1$), the variance simplifies to:

$$\text{Var}(w) = \frac{1}{N} \sum_{i=1}^{N} (w_i - 1)^2$$

Expanding this:

$$\text{Var}(w) = \frac{1}{N} \left( \sum_{i=1}^{N} w_i^2 - 2 \sum_{i=1}^{N} w_i + \sum_{i=1}^{N} 1 \right) = \frac{1}{N} \left( \sum_{i=1}^{N} w_i^2 - 2N + N \right) = \frac{1}{N} \left( \sum_{i=1}^{N} w_i^2 - N \right)$$

Rearranging to solve for $\sum w_i^2$:

$$\sum_{i=1}^{N} w_i^2 = N(1 + \text{Var}(w))$$

5. Recall the original formula:

$$n_{\text{eff}} = \frac{n^2}{\sum_{i=1}^{n} w_i^2}$$

Substitute $\sum w_i^2 = n(1 + \text{Var}(w))$:

$$n_{\text{eff}} = \frac{n^2}{n(1 + \text{Var}(w))} = \frac{n}{1 + \text{Var}(w)}$$

Thus, we obtain:

$$n_{\text{eff}} = \frac{n}{1 + \text{Var} \left( \frac{q(x)}{p(x)} \right)}$$

$\square$

Let $\nu(s, a)$ denote the state-action distribution of the dataset. In our setting, $p(s, a) = \nu(s, a)$, $q(s, a) \propto \nu(s, a) \exp(\beta(\hat{Q}(s, a) - \hat{V}(s)))$. Then

$$\text{Var}(p/q) = \text{Var}\left( \frac{\exp(\beta(\hat{Q}(s, a) - \hat{V}(s)))}{Z(s)} \right) = \hat{Q}_{\text{Var}}(\beta),$$

where $Z(s)$ is the normalization factor. $\hat{Q}_{\text{Var}}(\beta)$ is related to $\hat{Q}$ $\hat{V}$ and $\beta$. Use $n_{\text{eff}}$ to replace $n$ in Theorem A.1, we conclude our proof.

## B  EXPERIMENTAL DETAILS

All experiments are conducted on AMD EPYC$^{\text{TM}}$ 9654 CPUs and NVIDIA RTX 4090 GPUs. GPT-4o is used as our VLMs, while prompt designs are listed in Sec. B.3. The input to the VQ model can be a state representation $\phi(s)$ or a high-dimensional pixel-based input, but for the VLM, rendered observations are required.

### B.1  ENVIRONMENT DESCRIPTION

**Franka Kitchen**

Franka Kitchen (Gupta et al., 2019) offers a variety of tasks in a simulated kitchen environment, such as turning on the stove, turning on the light, opening microwave doors, and sliding drawers. Each collected trajectory sequentially lists these subtasks, making it well-suited for continuous control studies. The environment simulates realistic physical interactions, allowing the robot arm to perform complex actions like grasping, rotating, and sliding. With tasks naturally broken down into subtasks, it provides an ideal platform for exploring hierarchical reinforcement learning. Additionally, the sparse reward mechanism, where rewards are granted only upon completing specific goals, requires precise credit assignment. **Minigrid**

MiniGrid (Chevalier-Boisvert et al., 2023) is a lightweight, fast grid-world environment suite with OpenAI Gym interfaces, specifically designed for reinforcement learning tasks. Each environment consists of a 2D grid, where cells may be empty or occupied by objects such as walls, keys, doors, or goals. In our experiments, we fix both the initial and terminal states, using rendered images to represent the environment state, providing a controlled setting for reinforcement learning studies. Agents are required to perform actions like turning left or right, moving forward or backward, and interacting with objects such as picking up or dropping items to complete tasks efficiently. The reward function is sparse, with non-zero rewards typically given only upon task completion, and penalties applied based on the number of time steps taken.

**Crafter**

Crafter (Hafner, 2021) designs a survival game benchmark to evaluate reinforcement learning agents. Inspired by Minecraft, the game features procedurally generated 2D environments—forests, lakes, mountains, and caves—where agents perform survival tasks such as foraging, gathering resources, crafting tools, and defending against enemies. These tasks are organized into 22 semantically meaningful achievements, making Crafter an ideal environment for researching semantic skill extraction. Each new achievement unlocked during an episode rewards the agent with +1, making achievements the primary source of rewards in the game.

### B.2  MODEL ARCHITECTURE AND HYPERPARAMETERS

For the visual encoder, we adopt the same architecture as described in Liang et al. (2024). For the state encoder, we use a linear layer, and for the policy decoder, we employ an MLP network that maps the observation and skill context to the corresponding action. We list the hyperparameters used in VanTA for Kitchen, MiniGrid, and Crafter, respectively.

### B.3  PROMPT FOR VLMS

We use GPT-4o as the VLM to extract semantic skills. The query and response are separated into two parts: first, we provide the initial state, followed by the terminal state with a designed

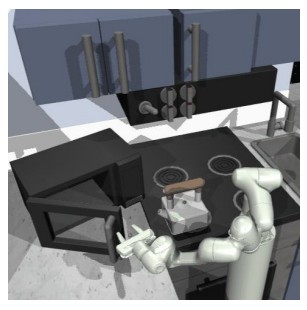
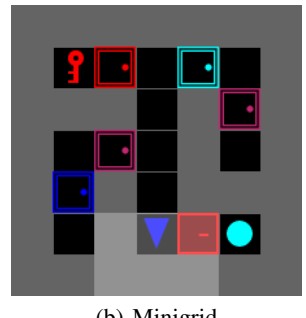
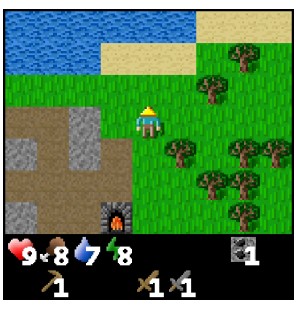

| (a) Franka Kitchen | (b) Minigrid | (c) Crafter |

Figure 5: Rendering of the procedurally-generated environments in our experiments.

Table 4: Hyperparameters and their values in VanTA.

| Hyperparameter | Kitchen | MiniGrid | Crafter |
|---|---|---|---|
| Batch size | 256 | 256 | 256 |
| Learning rate | 1e-4 | 1e-4 | 1e-4 |
| Optimizer | Adam | Adam | Adam |
| Iteration interval ($I$) | 50 | 20 | 20 |
| VQ option dim | 16 | 16 | 16 |
| VQ option num | 20 | 6 | 20 |
| $\beta$ | 3 | 5 | 5 |
| $\gamma$ | 0.1 | 0.2 | 0.2 |
| $\lambda$ | 0.99 | 0.99 | 0.99 |
| Gradient step | 3M | 1M | 2M |

prompt. The query format follows a user-assistant-user structure. This simulated dialogue significantly reduces VLM hallucinations in skill identification. However, the environment context differs slightly. For MiniGrid, the listed prompt is tailored to MiniGrid-RedBlueDoors, while for MiniGrid-KeyCorridorS3R3 and MiniGrid-DoorKey, the task descriptions require minor modifications.

**Prompt Design and Q&A procedure for kitchen**

**User:** Are you familiar with the Franka Kitchen environment used for reinforcement learning? In this environment, various skills represent different subtasks. You can deduce which skill was executed or identify the agent's subprocess based on the initial and terminal observations. I will now show you an image of the initial state. Here are examples of possible subtasks already summarized: 1. No obvious skill 2. Red flame circles appear on the stove as it turns on the gas stove 3. Rotate the knob to turn on the white light above the stove, which is the left one among those knobs 4. Open the cabinet 5. Move the kettle 6. Open the microwave. From my perspective, you can reason what the skill is according to the initial observation and terminal observation. Now I will show you the image, it is the initial state. If you think the task happened between two states in the summarized list, point the index out. Remember maybe many changes happen between two images, so choose the most suitable skill name. If not in them, and you still regard it as a new skill, try your best to give only one suitable new name. Each time, you can provide only one new name if necessary and carefully ensure that it is truly different from the names in the original set. In total, you can describe up to 20 skills.
**Assistant:** Got it! Please provide the terminal observation image, and I can help identify the skill or subtask executed between the initial and terminal states.
**User:** Now I will show you the image of the terminal state, which means it is the last frame of this segment. You should first judge what is the most obvious one if many changes happen. Check the difference and choose the most probable one. Please determine whether to choose from the given subtask list or name yourself. Write down your choice in the following format: the index selected by you; all the skills index and content. For example 1; 1.content of 1 2.content of 2 etc. Do not need to display your reasoning process.

**Prompt Design and Q&A procedure for MiniGrid**

**User:** Are you familiar with the Minigrid environment used for reinforcement learning? In this environment, Door Key task have several subtasks. This environment has a key that the agent must pick up in order to unlock a door and then get to the green goal square. You can deduce which skill was executed or identify the agent's subprocess based on the initial and terminal observations. I will now show you an image of the initial state. Here are all the possible subtasks already summarized: 1. In the middle part, move to the blue door 2. Try to get out from the left part 3. In the middle part, move to the red door 4. Reach the Goal. From my perspective you can reason what the skill is according to the initial observation and terminal observation. Now i will show you the image, it is the initial state.
**Assistant:** Got it, let's proceed with identifying the skills based on the initial and terminal observations. Please go ahead and show me the terminal observation so we can determine which subtask was executed or describe a new possible skill if necessary.
**User:** Now I will show you the image of the terminal state, which means it is the last frame of this segment. You should first judge what is the most obvious one if many changes happen. Please determine which one to choose from the given subtask list or name yourself. Write down your choice in the following format: the index selected by you; all the skills index and content i provided. For example 1;1. No obvious skill 2. Initial Positioning 3. Move to Key 4. Pass the door 5. Reach the goal position. Do not need to display your reasoning process.

**Prompt Design for Crafter**

> **User:** Are you familiar with the Crafter environment used for reinforcement learning? Crafter features randomly generated 2D worlds where the player needs to forage for food and water, find shelter to sleep, defend against monsters, collect materials, and build tools. It is the open world survival game for evaluating a wide range of agent abilities. The agent's view evolves as the timestep increases, like from wood to water. You can deduce which skill was executed or identify the agent's subprocess based on the initial and terminal observations. I will now show you an image of the initial state. Here are all the possible subtasks already summarized: 1. No obvious skill 2. collect sapling 3. collect stone. pay attention that the legend at the botton of the image shows new tools the agent made if it changes. From my perspective, you can reason what the skill is according to the initial observation and terminal observation. Now i will show you the image, it is the initial state. The task happened between two states is summarized in the list, just point out the index. Try your best to suspect the skill.
> **Assistant:** Thank you for sharing the initial state image. To proceed, please provide the terminal state image as well. With both the initial and terminal observations, I can help deduce the executed skill or identify the agent's subprocess according to the summarized list of possible subtasks.
> **User:** Now I will show you the image of the terminal state, which means it is the last frame of this segment. You should first judge what is the most obvious one if many changes happen. Please determine which one to choose from the given subtask list. Please only choose no obvious skill when indeed necessary, otherwise try your best to analyze according to the iconic item, like water, stone or wood! Write down your choice with the following format: the index selected by you; all the 16 skills index and content i provided. This is an example: 3; 1. No obvious skill 2. collect sapling 3. collect stone etc. Do not need to display your reasoning process.

### B.4 Pseudo-code for VanTA

We provide the pseudo-code for VanTA to further clarify the process. During the offline initialization phase, we iterate at 2 times the VLM frequency to prepare the initial segmentation and reduce query complexity in the early stages. The absence of smooth loss may give rise to misleading initialization which is not conducive for VLM to reidentify the sub-task. Therefore, we add the smooth loss with the appropriate coefficient, if meeting the no obvious skill option, the coefficient will increase to 10 times.

---

**Algorithm 1** Vision-language model guided Temporal Abstraction (VanTA) for offline RL

---

**Require:** offline data $\mathcal{D}$; pretrained VLM; high-level policy $\pi_\theta$; low-level policy $\pi_\phi$;
        query VLM frequency $I$; temporal abstraction epoch $N$; policy update epoch $M$.
1: // Primitive Skill Extraction
2: **for** $iter = 1$ to $N$ **do**
3:     Sample a minibatch $\{(s, a, s', r)\}$ from $\mathcal{D}$
4:     Update *codebook* with Eq. (3)
5:     **if** step meets query VLM frequency $I$ **then**
6:         Query VLM with initial segmentation $\{\bar{s}_1, \bar{s}_2, ...\}$
7:         Update *codebook* with VLM indice using Eq. (2)
8:     **end if**
9: **end for**
10: Relabel offline data with primitive skill as $\mathcal{D}^z = \{(s, a, s', r, z)\}$
11: // Policy Extraction
12: $\pi_\omega, \pi_\theta = $ Policy Extraction$(\mathcal{D}^z, M)$ .
**Output:** policy $\pi_\omega, \pi_\theta$.

---

## B.5 ADDITIONAL RESULTS

### B.5.1 MISSING SKILL EXTRACTION RESULTS

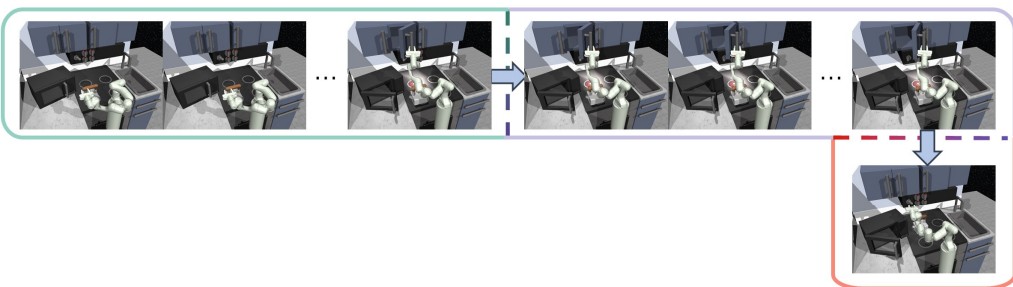

Figure 6: Skill extraction results without VLM guidance in kitchen-mixed; no clear semantic meaning in each skill.

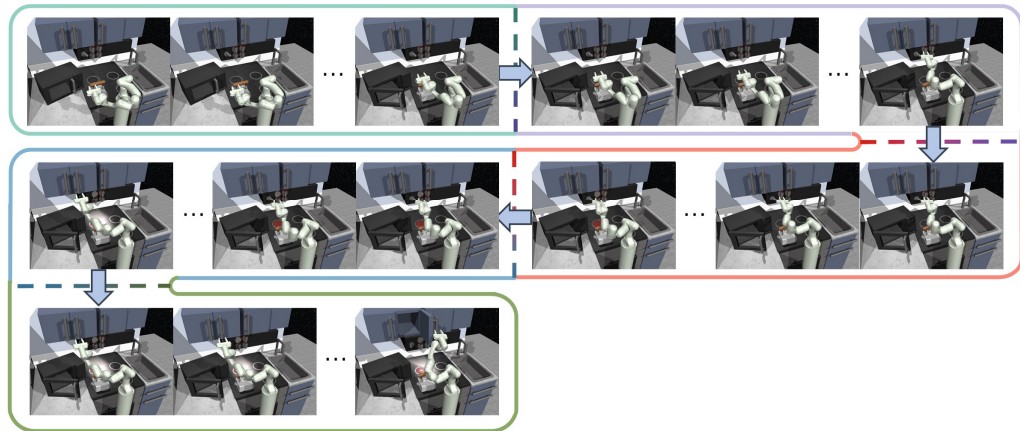

Figure 7: Skill extraction results using VanTA in kitchen-mixed: sequentially **1) open the microwave, 2) red flame circles appear on stove as it turns on the gas stove, 3) rotate the knob to turn on the white light above the stove, which is the left one among those knobs, and 4) open the cabinet**.

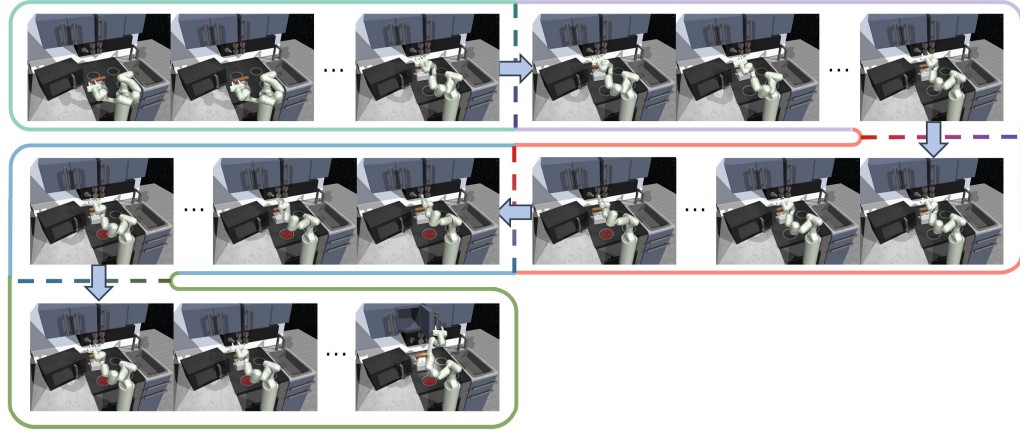

Figure 8: Skill extraction results without VLM guidance in kitchen-partial; no clear semantic meaning in each skill.

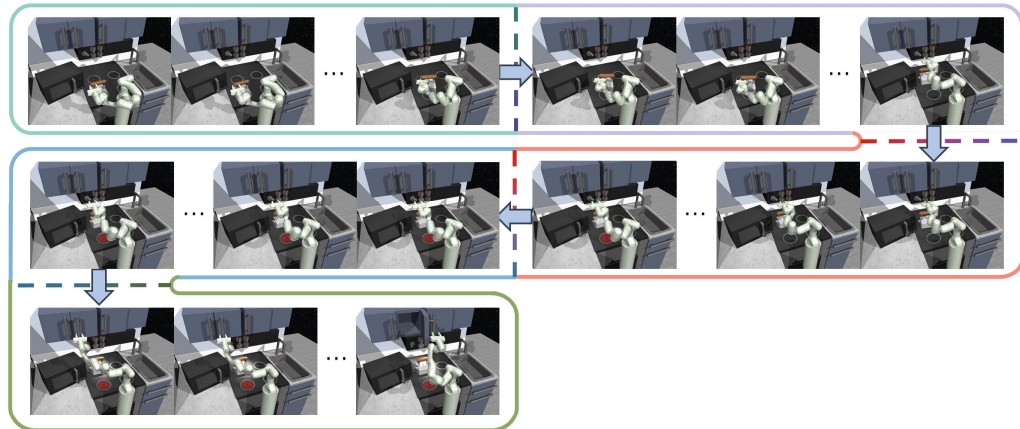

Figure 9: Skill extraction results using VanTA in kitchen-partial: sequentially **1) Move the kettle, 2) red flame circles appear on the stove as it turns on the gas stove, 3) rotate the knob to turn on the white light above the stove, which is the left one among those knobs, and 4) open the cabinet**.

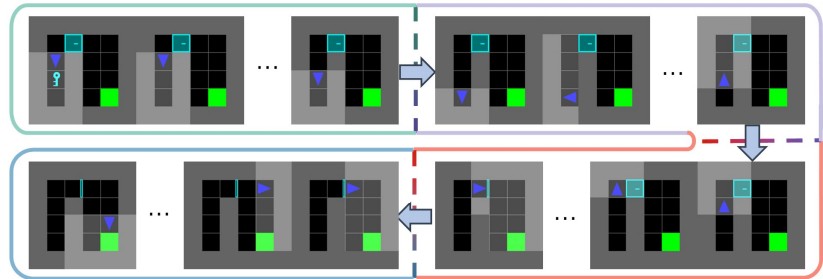

Figure 10: Skill extraction results without VLM guidance in MiniGrid-DoorKey-6×6; the baseline method mistakenly treats wandering as a valid action.

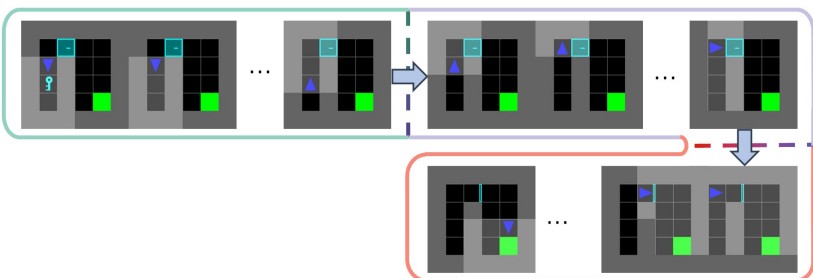

Figure 11: Skill extraction results using VanTA in MiniGrid-DoorKey-6×6: sequentially **1) move to key, 2) pass the door, and 3) reach the goal position**. Our method accurately segments the dataset into these three essential skills without any redundant parts.

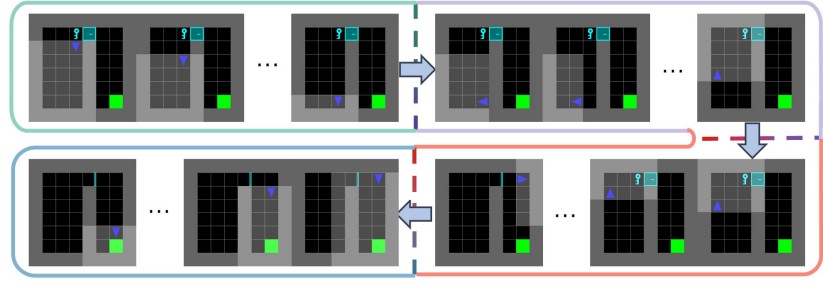

Figure 12: Skill extraction results without VLM guidance in MiniGrid-DoorKey-8×8; no clear semantic meaning in each skill.

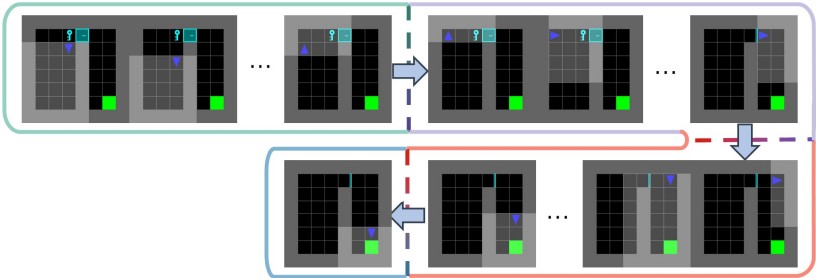

Figure 13: Skill extraction results using VanTA in MiniGrid-DoorKey-8×8: sequentially **1) move to key, 2) pass the door, and 3) reach the goal position**. Unlike the baseline method, our approach can effectively separate the semantic information for 'move to the key' and 'pass the door'.

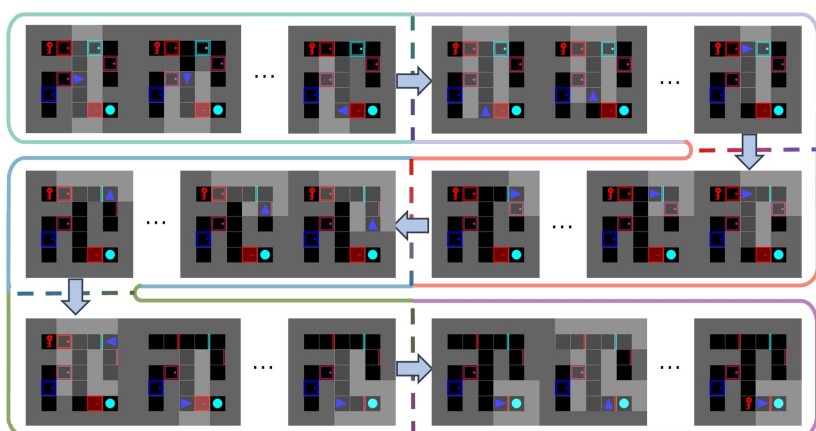

Figure 14: Skill extraction results without VLM guidance in MiniGrid-KeyCorridorS3R3; although it can be roughly divided into four semantic parts: move to the sky blue door at the top right corner, move to the purplish red door at the top right corner, reach the red key at the top left corner, and move to the red door at the bottom right corner, there are still some unnecessary skills in between.

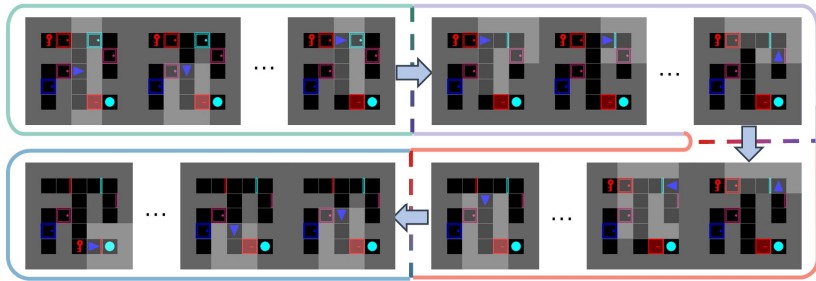

Figure 15: Skill extraction results using VanTA in MiniGrid-KeyCorridorS3R3: sequentially **1) move to the sky blue door at the top right corner, 2) move to the purplish red door at the top right corner, 3) reach the red key at the top left corner, and 4) move to the red door at the bottom right corner**. Compared to the baseline, our method merges segments that do not carry meaningful semantics.

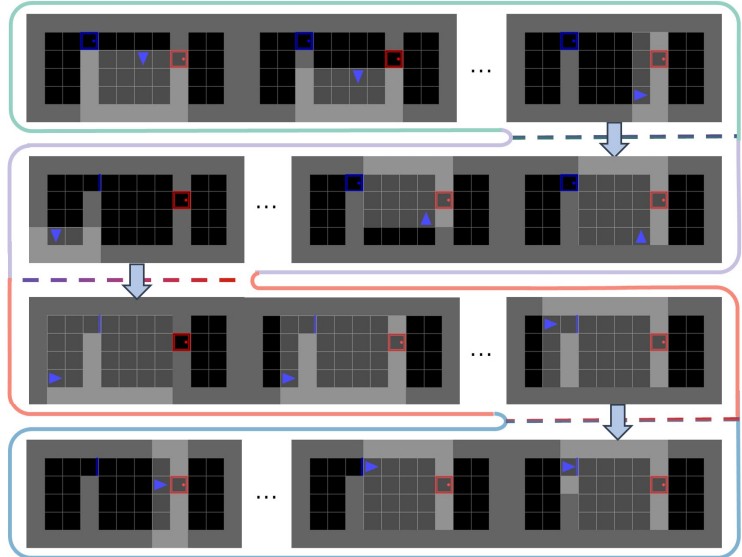

Figure 16: Skill extraction results without VLM guidance in MiniGrid-RedBlueDoors; no clear semantic meaning in each skill.

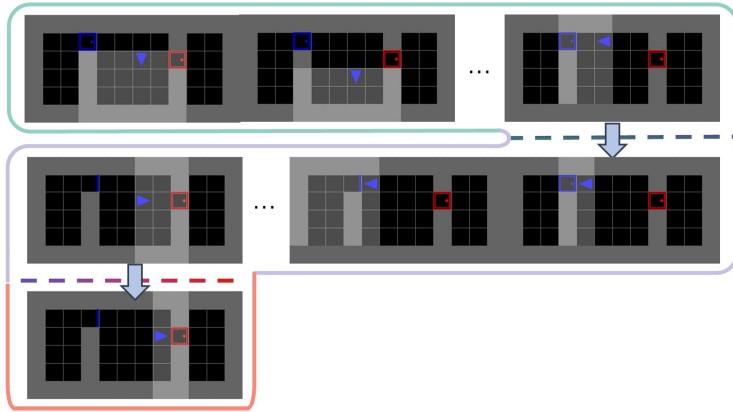

Figure 17: Skill extraction results using VanTA in MiniGrid-RedBlueDoors: sequentially **1) in the middle part move to the blue doors, and 2) in the middle part move to the red door**.

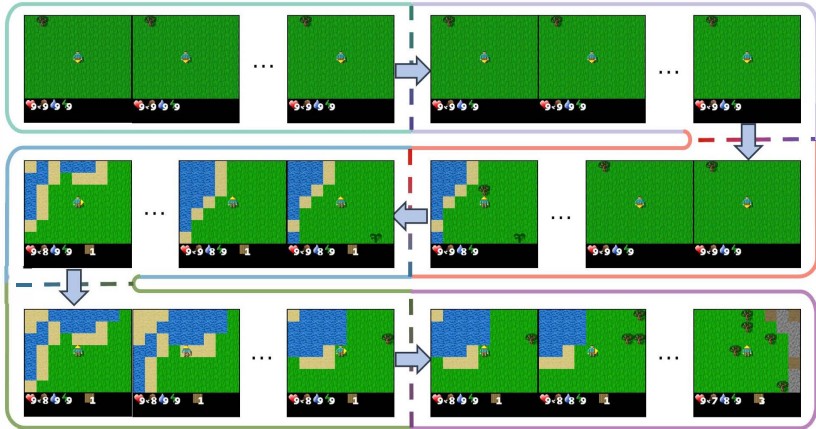

Figure 18: Skill extraction results without VLM guidance in Crafter-partial; no clear semantic meaning in each skill.

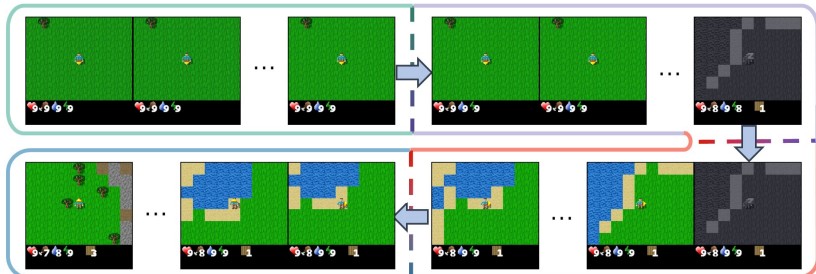

Figure 19: Skill extraction results using VanTA in Crafter-partial: sequentially **1) collect sapling, 2) sleep to refresh, 3) collect water to drink, and 4) collect wood**.The extracted segments largely correspond to the achievements in the Crafter environment, and there are significant scene differences between distinct segments.

### B.5.2 VISUALIZATION OF THE REDUCED DATA EXPERIMENTS

We have included a graphical visualization that more clearly demonstrates the slower degradation of VanTA compared to other methods, such as CQL and GCSL. We believe this visualization highlights the comparison more effectively and strengthens the clarity of our claims. The graphical representation is provided in Fig. 20.

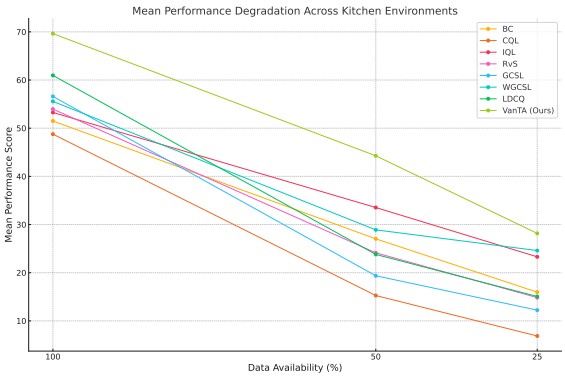

Figure 20: Visualization of reduced offline data experiment.

### B.5.3 FULL ABLATION STUDY

We conduct an ablation study across all environments, comparing the performance of VanTA with its variant that lacks VLM guidance. The results show that conditioning training on irrational primitive skills negatively impacts the final policy performance.

Table 5: Ablation study on VanTA and conditioning only non-VLM guidance skills, averaged over 5 random seeds.

| Task Name | w/o VLM Guidance | w/ VLM Guidance |
|---|---|---|
| MiniGrid-DoorKey-6x6-complete-v0 | 0.91±0.02 | **0.92±0.03** |
| MiniGrid-DoorKey-6x6-mixed-v0 | 0.72±0.14 | **0.80±0.17** |
| MiniGrid-DoorKey-8x8-complete-v0 | 0.90±0.03 | **0.92±0.07** |
| MiniGrid-DoorKey-8x8-mixed-v0 | 0.43±0.14 | **0.51±0.18** |
| MiniGrid-KeyCorridorS3R3-complete-v0 | 0.77±0.12 | **0.81±0.06** |
| MiniGrid-KeyCorridorS3R3-mixed-v0 | 0.47±0.09 | **0.61±0.12** |
| MiniGrid-RedBlueDoors-6x6-complete-v0 | 0.82±0.07 | **0.85±0.11** |
| MiniGrid-RedBlueDoors-6x6-mixed-v0 | 0.67±0.12 | **0.73±0.20** |
| Crafter-partial | 2.8 | **5.46** |

### B.5.4 DETAILS OF CRAFTER PERFORMANCE

We observe that the policy induced by VanTA achieves the highest success rate in almost half of the achievements. The overall score is calculated as the geometric mean of success rates across all achievements.

Table 6: Performance listing of all achievements on Crafter benchmark for VanTA and 5 baselines, averaged over 5 random seeds.

| Achievement | Success Rate (%) | | | | | |
|---|---|---|---|---|---|---|
| | BC | CQL | IQL | RvS | LDCQ | VanTA(Ours) |
| Collect Coal | 0.2 | 0.0 | 0.0 | 0.0 | 0.0 | **0.6 ± 0.2** |
| Collect Diamond | 0.0 | 0.0 | 0.0 | 0.0 | 0.0 | 0.0± 0.0 |
| Collect Drink | 15.6 | **33.4** | 4.2 | 20.0 | 0.0 | 17.9± 5.5 |
| Collect Iron | 0.0 | 0.0 | 0.0 | 0.0 | 0.0 | 0.0 ± 0.0 |
| Collect Sapling | 42.8 | 63.4 | 88.2 | 90.6 | **99.0** | 78.7± 9.4 |
| Collect Stone | 2.4 | 0.0 | 0.0 | 0.0 | 0.0 | **4.4±2.2** |
| Collect Wood | 73.6 | 53.8 | 60.2 | 73.8 | 0.0 | **74.1±8.5** |
| Defeat Skeleton | 0.2 | 0.4 | 1.8 | 0.0 | 0.0 | **2.2 ± 0.7** |
| Defeat Zombie | 11.8 | 6.2 | 20.8 | **28.2** | 15.0 | 21.8±6.3 |
| Eat Cow | 21.6 | 7.4 | 27.0 | 20.0 | 15.6 | **27.2±2.4** |
| Eat Plant | 0.1 | 0.0 | **0.2** | 0.0 | 0.0 | 0.0 ± 0.0 |
| Make Iron Pickaxe | 0.0 | 0.0 | 0.0 | 0.0 | 0.0 | 0.0± 0.0 |
| Make Iron Sword | 0.0 | 0.0 | 0.0 | 0.0 | 0.0 | 0.0± 0.0 |
| Make Stone Pickaxe | 0.0 | 0.0 | 0.0 | 0.0 | 0.0 | 0.0 ± 0.0 |
| Make Stone Sword | 0.0 | 0.0 | 0.0 | 0.0 | 0.0 | 0.0 ± 0.0 |
| Make Wood Pickaxe | **31.0** | 0.0 | 0.4 | 0.4 | 0.0 | 30.5± 6.7 |
| Make Wood Sword | 17.2 | 0.0 | 15.0 | 0.0 | 0.0 | **21.7±8.9** |
| Place Furnace | 0.0 | 0.0 | 0.0 | 0.0 | 0.0 | 0.0±0.0 |
| Place Plant | 0.0 | 7.2 | 86.8 | 70.6 | **98.0** | 64.4± 5.6 |
| Place Stone | 0.0 | 0.0 | 0.0 | 0.0 | 0.0 | **4.4±1.2** |
| Place Table | 53.8 | 0.0 | 36.8 | 7.0 | 0.0 | **73.1 ± 2.4** |
| Wake Up | 0.2 | **47.6** | 0.0 | 0.0 | 0.0 | 16.7±8.1 |
| Score | 2.69 | 1.73 | 2.75 | 2.11 | 0.96 | **5.46** |

*The score is computed as $S = \exp\left(\frac{1}{N}\sum_{i=1}^{N}\ln(1+s_i)\right) - 1$, where $s_i \in [0, 100]$ is the success rate of the $i$-th achievement and $N = 22$ denotes the total number of achievements.

Table 7: Performance listing of all achievements for VanTA ablation study on Crafter benchmark.

| Achievement | Success Rate (%) | |
|---|---|---|
| | w/o VLM Guidance | w/ VLM Guidance |
| Collect Coal | 0.0± 0.0 | **0.6 ± 0.2** |
| Collect Diamond | 0.0± 0.0 | 0.0± 0.0 |
| Collect Drink | 10.8±6.0 | **17.9± 5.5** |
| Collect Iron | 0.0±0.0 | 0.0 ± 0.0 |
| Collect Sapling | **94.2± 2.2** | 78.7± 9.4 |
| Collect Stone | 0.0±0.0 | **4.4±2.2** |
| Collect Wood | 41.3±16.3 | **74.1±8.5** |
| Defeat Skeleton | 1.1±0.7 | **2.2 ± 0.7** |
| Defeat Zombie | 20.3 ± 2.3 | **21.8±6.3** |
| Eat Cow | 20.5 ± 2.0 | **27.2±2.4** |
| Eat Plant | 0.0 ± 0.0 | 0.0 ± 0.0 |
| Make Iron Pickaxe | 0.0 ± 0.0 | 0.0± 0.0 |
| Make Iron Sword | 0.0 ± 0.0 | 0.0± 0.0 |
| Make Stone Pickaxe | 0.0 ± 0.0 | 0.0 ± 0.0 |
| Make Stone Sword | 0.0 ± 0.0 | 0.0 ± 0.0 |
| Make Wood Pickaxe | 3.2 ± 1.7 | **30.5± 6.7** |
| Make Wood Sword | 2.5 ± 1.6 | **21.7±8.9** |
| Place Furnace | 0.0±0.0 | 0.0±0.0 |
| Place Plant | **81.5±9.4** | 64.4± 5.6 |
| Place Stone | 1.6± 1.2 | **4.4±1.2** |
| Place Table | 18.2±1.8 | **73.1 ± 2.4** |
| Wake Up | 1.8±1.6 | **16.7±8.1** |
| Score | 2.85 | **5.46** |

*The score is computed as $S = \exp\left(\frac{1}{N}\sum_{i=1}^{N}\ln(1+s_i)\right) - 1$, where $s_i \in [0, 100]$ is the success rate of the $i$-th achievement and $N = 22$ denotes the total number of achievements.

