# OpenReview forum: "Semantic Temporal Abstraction via Vision-Language Model Guidance for Efficient Reinforcement Learning"
_ICLR.cc/2025/Conference — ICLR 2025 Poster_

### Official Review · Reviewer_R8wN · 2024-10-29

**Soundness:** 3
**Presentation:** 3
**Contribution:** 3
**Rating:** 6
**Confidence:** 4

**Summary:**

This paper introduces VanTA, an approach that integrates vision-language models (VLMs) into offline reinforcement learning (RL) for sparse reward settings. The author presents a hierarchical offline RL framework, where high-level skill policy training is augmented by the use of VLMs. Specifically, VLM is queried with image pairs to identify performed skills (e.g., pulling, pushing, picking, …), and this result is incorporated into the learning process of vector-quantized variational autoencoder (VQVAE), aiming to guide the discrete latent skill space to be more semantically meaningful. Conditioned on the skill, low-level control policy is separately learned via behavior cloning. Experiments on multiple environments show the superiority of VanTA.

**Strengths:**

1. The research direction of grounding VLMs to manipulation and beyond represents a significant and promising area of investigation in the field.
2. The paper is well-organized and demonstrates consistent terminology usage, making it clear to readers.
3. The experimental validation is comprehensive, spanning multiple environments.

**Weaknesses:**

1. The primary concern is the novelty of the contribution. The concept of extracting semantic skills via VLMs and grounding them to action policies has been previously explored, as demonstrated in [1]. The authors should clearly differentiate their approach from existing work, particularly regarding their VQVAE implementation compared to [1].

2. A fundamental aspect of skill-based RL approaches is their ability to reuse extracted skills while reducing action space exploration, enabling rapid [2, 3] or even zero-shot [1] adaptation to new tasks. The paper should elaborate on the method's generalization capabilities beyond the tasks represented in the offline dataset.

3. The application of VLMs in this work may not fully leverage their potential. Foundation models' primary advantage lies in their domain-agnostic knowledge, which should facilitate rapid policy adaptation across diverse domains. However, the proposed method's reliance on domain-specific reward structures potentially limits its generalization capabilities. The authors should address whether policies trained on specific tasks can generalize to different tasks.

[1] One-shot Imitation in a Non-Stationary Environment via Multi-Modal Skill, ICML 2023

[2] Skill-Based Meta-Reinforcement Learning, ICLR 2022

[3] Accelerating reinforcement learning with learned skill priors, CoRL 2020

**Questions:**

1. The implementation of behavior cloning (BC) for low-level policy learning (Line 229) appears to require expert demonstrations in the offline dataset. Then, how does BC-based learning achieve good performance with mixed-quality datasets in the Minigrid experiments?

2. When reading through the middle sections of the paper, I  expect the VQVAE's encoder and decoder to serve as the high-level and low-level policies, respectively. Given that the training already utilizes expert datasets and Line 159 indicates that the VQVAE's decoder is replaced with a policy decoder, the authors should justify their architectural decision to train separate high-level and low-level policies instead of leveraging the VQVAE's existing structure.

3. Given the large model size of the VLM used in this paper, I am not surprised that it outperforms other baselines. Did other baselines employ models of comparable size to the VLM?

4. Was there any specific fine-tuning or prompt engineering applied to the VLM? While Line 104 points out the limited reasoning abilities of VLMs, it appears that this paper uses them as-is without any specialized adaptation techniques.

---

> ### Author Response · Authors · 2024-11-22
>
> 1. Novelty of the contribution.
>
> **Response:**
>   We understand the reviewer's concern regarding the novelty of our contribution and would like to address it clearly.
>
>   We aim to solve a complex single task with sparse reward and long-horizon, it usually can be decomposed sequentially into subtasks or subgoals. In our framework, a skill is defined as a contextual policy, where context is a signal representing a subtask or subgoal, providing temporally extended guidance that helps the agent navigate complex tasks. To achieve this, we combine VQ with a Vision-Language Model (VLM), which provides semantic guidance to shape the latent space of VQ (skill context representations). The VLM progressively guides the learning of the codebook, enabling the discovery of skills without the need for additional annotations, such as language instructions.
>
> Here, we emphasize that the key contribution of our work lies in introducing an innovative approach to grounding the latent space of VQ with semantic meaning for decision-making tasks, which is fully automated.
>
> In contrast, previous work, such as [1], relies heavily on human-labeled subtask-level or episode-level language descriptions, where language definitions serve as the ground truth for discovering skills.
>   Besides, [1] assumes fixed-length segmentation, which encodes actions within a fixed horizon $t + H$ into a skill context. This fixed segmentation approach is inherently limited, as it fails to adapt to primitive actions of varying lengths, often resulting in inaccurate task decomposition.
> Our method, however, automatically extracts the start and end timesteps corresponding to the same skill context, alleviating issues caused by misleading information from predefined skill horizons. This approach enables more flexible skill discovery, particularly in scenarios where the horizon of each primitive action varies.
>
> [1] Shin S, Lee D, Yoo M, et al. One-shot imitation in a non-stationary environment via multi-modal skill.
>
>
> 2. Skills should be able to generalize beyond the offline dataset.
>
> **Response:** Thanks for the question. As clarified before, our primary focus is on a setting involving a single task composed of sequential subtasks, characterized by long horizons and sparse rewards. Our skill context learning is designed to identify a temporally extended signal representing the current subtask or the next subgoal to reach. As such, in the original manuscript, we did not explicitly explore the generalization capabilities of the learned "skill," as our method was tailored for this single task structure.
>
> However, we also appreciate the importance of generalization and have conducted additional experiments to evaluate our method’s performance in this regard. To achieve this, we train a task embedding to identify tasks in unseen settings. Initially, we train our VanTA algorithm in a multi-task environment to extract reusable skills from task-agnostic offline data. These learned skills serve as priors that can be adapted to new tasks. Subsequently, we train a high-level policy conditioned on the task embedding to select optimal skills, with the priors from offline data acting as constraints. Here, the low-level policy learning just aims to extract the skill, without offline rl optimization.
> Due to limited time, we conduct experiments only in the Franka Kitchen environment. The results are available at [https://imgur.com/a/r1g3V9C](https://imgur.com/a/r1g3V9C), where unseen tasks consist of novel subtask sequences. Our observations show that our method outperforms PEARL, MTRL, and SAC in manipulation tasks, demonstrating its generalization capabilities. Generalization in other scenarios will be explored in future work.
>
> 3. The application of VLMs in this work may not fully leverage their potential.
>
> **Response:** VLMs are powerful tools as they excel in understanding and reasoning with both language and images. In our method, we leverage VLMs to extract semantically meaningful skill contexts, which serve as a context for low-level policy learning. It is important to note that the skill extraction process itself does not rely on domain-specific reward structures, as the VLM labels the task segments using its domain-agnostic knowledge.
>
> However, while VLMs effectively provide semantic guidance for high-level skill learning, the low-level policy, even with VLM integration, cannot be entirely universal. This is largely due to differences in action spaces across domains, as well as variations in task goals. These differences necessitate task-specific training for low-level policies to perform effectively.
> Despite this, we demonstrate that the learned skills facilitate a certain degree of generalization across tasks. We conducted experiments in the Franka Kitchen environment, where the learned skills were tested on unseen subtask sequences.

---

> ### Author Response · Authors · 2024-11-22
>
> 4. Why can VanTA handle mixed-quality datasets?
>
> **Response:**  Even though the  VQ-VAE encoder and decoder have been trained as high-level and low-level policies, respectively, we continue to use the encoder to map states to skill embedding. However, the decoder is replaced with a low-level policy and re-trained with IQL following the rule below:
> $$ \min_{\omega} J(\omega; \mathcal{D}) = \hat{\mathbb{E}}\_{\tau \sim \mathcal{D}} \left[ -\sum_{t=0}^{H-1} \exp(\zeta(Q(s_t, a_t) - V(s_t))) \log \pi_{\omega}(a_t | s_t, z_t) \right]. $$
> It allows for adaptation to varying dataset qualities by effectively leveraging the reward signals present in the offline data. This explains why our method performs well even in the presence of mixed-quality datasets.
>
> 5. Justify the architectural decision to train separate high-level and low-level policies instead of leveraging the VQVAE's existing structure.
>
> **Response:** Thank you for your question. While the VQ-VAE encoder and decoder could theoretically serve as high-level and low-level policies, respectively, we find that separating these components enhances performance. In our approach, the high-level policy continues to map states to actions, while the low-level policy is trained separately using IQL, which facilitates adaptation to datasets of varying quality.
>
> Additionally, the reason for using a Q-learning method to optimize the low-level policy stems from our main idea, which is to leverage VLM semantic guidance to build a better skill context for policy learning. This context serves as a signal representing the current subtask or the next subgoal to reach.
> Once the context is identified, skills can be further optimized using rewards to achieve better performance in tasks with sparse rewards or long horizons.
>
> 6. Did other baselines employ models of comparable size to the VLM?
>
> **Response:** We fully understand the reviewer’s concern.  While larger models may improve performance, simply increasing the model size does not necessarily lead to better results. For example, in subsequent experiments, we incrementally increase parameters (2$\times$ the width and 3$\times$ the depth, then 4$\times$ the width and 6$\times$ the depth) in both value networks and policy networks. However, this actually degrades performance, highlighting that increasing the model size alone is not always beneficial.
>
> | **Task Name**          | [256, 256]       | **2× width 3× depth** | **4× width 6× depth** |
> |-------------------------|------------------|------------------------|------------------------|
> | kitchen-complete-v0     | 69.2 ± 8.5      | 43.8 ± 4.2            | 28.5 ± 5.2            |
> | kitchen-partial-v0      | 71.2 ± 5.7      | 45.1 ± 4.7            | 17.3 ± 4.3            |
> | kitchen-mixed-v0        | 68.5 ± 4.4      | 52.2 ± 7.1            | 22.0 ± 6.1            |
>
> Furthermore, the VLM is pretrained on a large scale of data, effectively serving as a knowledge base to extract useful information. It acts as an external module and our main idea is to effectively leverage the external knowledge to learn better skill context. Therefore, we hold that it does not result in an unfair comparison in terms of model size.
>
> 7. Was there any specific fine-tuning or prompt engineering applied to the VLM?
>
> **Response:** The GPT-4o model used in our approach is a pre-trained foundation model that leverages internet-scale knowledge to accelerate the training process. We note that VLMs may have limited reasoning abilities when handling complex tasks, such as generating step-by-step low-level plans based on the current state or dividing an entire trajectory into sub-trajectories. Our work, however, takes a different approach. We utilize the VLM specifically for its identification ability to identify skills and, in addition, apply several technical tricks to further enhance its effectiveness.
>
> The core idea is to simplify input complexity to improve the VLM's reasoning capabilities. For example, we initialize the subgoal by setting it using the VQ classification result, which significantly enhances skill segmentation and discovery. Additionally, we employ role-aware prompt engineering: an initial image of the segment is sent to the VLM, its output is logged, and the message is then forwarded with appropriate role management (e.g., assigning an assistant role). This approach enables the VLM to recognize skill names based on terminal states. Detailed examples of our prompt engineering techniques are provided in the Appendix.
>
> We hope our response can properly address your concern and sincerely look forward to your feedback.

---

> > ### Comment · Reviewer_R8wN · 2024-11-24
> >
> > Thank you for your quick responses and the additional experiments provided within the limited time. While your rebuttal has addressed many of my initial concerns, leading me to raise my overall score, I remain uncertain whether the contribution meets ICLR's standards.
> >
> > First, the concept of extracting compositionality from long-horizon complex tasks using VLMs has been previously explored, and your differentiation from prior work appears to be largely implementation-based. While I acknowledge that your approach is more oriented towards unsupervised algorithms through automated semantic skill extraction, it's unclear whether existing approaches, when combined with recent VLMs, couldn't achieve similar automation without labels.
> >
> > Additionally, your focus on skill discovery rather than generalizability while using foundation models seems to fall into an awkward middle ground. On one side, the RL community has established methodologies for complete unsupervised skill discovery, and on the other, they have previous works that leverage VLMs to demonstrate generalization capabilities. Your work, while interesting, doesn't seem to fully commit to either direction, making its positioning and contribution less clear.

---

### Official Review · Reviewer_nVvm · 2024-11-01

**Soundness:** 2
**Presentation:** 2
**Contribution:** 2
**Rating:** 6
**Confidence:** 4

**Summary:**

This paper proposes a novel method called VanTA (Vision-language model guided Temporal Abstraction) that utilizes Vision-Language Models (VLMs) to extract meaningful skills from offline reinforcement learning. VanTA extracts skills through an iterative process that involves initial segmentation based on VQ-VAE, followed by using VLM to assign meaning to the skills and update the codebook.

**Strengths:**

The paper proposes a novel method called VanTA (Vision-language model guided Temporal Abstraction) that utilizes Vision-Language Models (VLMs) to extract meaningful skills from offline reinforcement learning. It claims to overcome the limitations of existing unsupervised learning approaches or methods that require human intervention.

**Weaknesses:**

The experiment section lacks baselines related to skill learning among the comparisons. It is difficult to determine whether VanTA's performance is due to the proposed algorithm or simply because it uses a skill learning framework. Given that skills are learned in the form of a codebook, a comparison with [1] seems necessary, and a comparison with [2], which utilizes LLM, also appears to be needed.

[1] Mazzaglia, Pietro, et al. "Choreographer: Learning and adapting skills in imagination." *arXiv preprint arXiv:2211.13350* (2022).
[2] Fu, Haotian, et al. "Language-guided Skill Learning with Temporal Variational Inference." *arXiv preprint arXiv:2402.16354* (2024).

**Questions:**

* The experiment section appears to lack baselines related to skill learning among the comparisons. It seems necessary to include comparison groups associated with skill learning in the baselines.

---

> ### Author Response · Authors · 2024-11-22
>
> 1. It seems necessary to include comparison groups associated with skill learning in the baselines.
>
> **Response:** We appreciate your valuable comment regarding the need for skill learning baselines. Our experiments have included a comparison with a non-VLM guidance method, which is our ablation study. This aims to demonstrate that simply using a skill learning framework is not sufficient to achieve the performance improvements observed with VanTA. Additionally, we compare our approach to LDCQ, a state-of-the-art method that utilizes a skill learning framework with constraints to encode trajectory sequences into compressed latent skills. This comparison further highlights the advantages of VanTA.
> Regarding the baseline comparison with [1], we have included this method in our experiments as follows.
> | **Task Name**                            | **Choreographer**   | **VanTA**            |
> |------------------------------------------|---------------------|----------------------|
> | kitchen-complete-v0                      | 53.2 ± 4.2          | **69.2 ± 8.5**       |
> | kitchen-partial-v0                       | 60.4 ± 5.2          | **71.2 ± 5.7**       |
> | kitchen-mixed-v0                         | 62.5 ± 5.9          | **68.5 ± 4.4**       |
> | MiniGrid-DoorKey-6x6-complete-v0         | **0.93 ± 0.03**     | 0.92 ± 0.03          |
> | MiniGrid-DoorKey-6x6-mixed-v0            | 0.72 ± 0.14         | **0.80 ± 0.17**      |
> | MiniGrid-DoorKey-8x8-complete-v0         | 0.89 ± 0.10         | **0.92 ± 0.07**      |
> | MiniGrid-DoorKey-8x8-mixed-v0            | 0.48 ± 0.11         | **0.51 ± 0.18**      |
> | MiniGrid-KeyCorridorS3R3-complete-v0     | 0.75 ± 0.07         | **0.81 ± 0.06**      |
> | MiniGrid-KeyCorridorS3R3-mixed-v0        | 0.44 ± 0.08         | **0.61 ± 0.12**      |
> | MiniGrid-RedBlueDoors-6x6-complete-v0    | 0.84 ± 0.04         | **0.85 ± 0.11**      |
> | MiniGrid-RedBlueDoors-6x6-mixed-v0       | 0.65 ± 0.06         | **0.73 ± 0.20**      |
> | Crafter-partial                          | 2.98                | **5.46**             |
>
> As described in [1], the method supports both state-based and visual-based settings. For the Kitchen environment, we utilized the state-based deployment code provided in [1]. Similarly, for the MiniGrid and Crafter environments, which rely on visual observations, we employed the visual-based deployment code as specified in the original paper.
> As our approach assumes no additional exploratory data, we trained the model solely on the available dataset for fair comparison. We discover that the intrinsic rewards for the low-level skill-conditioned policy rely on the rollout model, and the performance becomes sensitive to parameters such as the rollout horizon, potentially introducing compounding errors.
>
> We also attempt to compare our approach with [2]. However, this method assumes that every action has a semantic meaning, which is described by language. For instance, actions are annotated with descriptions such as "move ahead," along with details like the distance to move. In contrast, our approach does not rely on such human annotations. Instead, we aim to autonomously extract skills directly from the observations, with the VLM independently discovering the skills without requiring predefined semantic labels or action descriptions.
>
> We hope these clarifications address the reviewer’s concern and sincerely look forward to the feedback.
>
> [1] Mazzaglia P, Verbelen T, Dhoedt B, et al. Choreographer: Learning and adapting skills in imagination.
>
> [2] Fu H, Sharma P, Stengel-Eskin E, et al. Language-guided Skill Learning with Temporal Variational Inference.

---

### Official Review · Reviewer_QKoU · 2024-11-03

**Soundness:** 3
**Presentation:** 2
**Contribution:** 3
**Rating:** 6
**Confidence:** 3

**Summary:**

The method proposes a hierarchical policy, with a high-level skill selection policy trained via offline RL. A second low-level policy trained by behavior cloning outputs the actions given the current state and selected skill. For the skill extraction, a vector quantizied VAE is trained with labels provided by a VLM.

**Strengths:**

- The proposed hierarchical policy is an interesting method, with promising results.
- The discretization offered by VQ-VAE enables the use of Q-Learning for the high-level skill policy while retaining the continuous output space of a low-level action policy.

**Weaknesses:**

- The clarity of the text should be improved. It often is hard to parse, and the text leaves it unclear, what the intentions are. For Example:
    - The introduction leaves it unclear what the authors define as "skill"
    - The Preliminary section's formatting makes it hard to parse. The Markov Decision Process (MDP) abbreviation is never actually defined. For VQ-VAE sg() is not defined. Same for Eq. 3
    - Section 5 suddenly appears without any stated goal or context. What is it supposed to show?
    The experiment section introduction is hard to read. In the last sentence of Section Five, the text describes the following order:
    6.5 -> 6.2 -> 6.1 -> 6.4 -> 6.3. Then, it references Fig. 5, which is late in the Appendix.
    - Line 210 states Eq. 3 but actually references Eq. 1
- Figure 1 suggests the policy extraction process requires the codebook model. But the High-level policy, in my understanding, outputs a skill selection, i.e., a codebook entry. Why does it require the codebook?
- Please clarify: Is the VLM only used during training or during rollout? Follow-up Question for the Ablation 6.3: For the method without VLM, in which stages has it been left out?
- The mentioned main limitation is "proper initialization". Please explain what initialization means, as this is barely mentioned in the rest of the main text.
- Ablations regarding the VAE policy decoder are missing. In general, the architecture of the policy decoder is not clear from the paper.

The proposed method and use of VQ-VAE + VLM is interesting. However, in its current state, the paper is hard to parse. Without improvements to the text, I tend towards reject.

**Questions:**

- Out of interest: How many skills were identified with VLM guidance compared to those without VLM guidance?
- From my understanding, the codebook is learned self-supervised with labels provided by the VLM. Do you have any experiments in a purely IL setting? Or could you provide further motivation for the policy extraction via Q-Learning dependent on a reward function?
- For 6.4: could you provide some graphical visualization of the table? I believe this would make the claimed slower degradation more apparent.
- Does the VQ get a single frame or multiple frames to segment? How does the segmentation work? How are the initial and terminal state of a primitive skill $\bar{s}$ determined?

---

> ### Author Response · Authors · 2024-11-22
>
> 1. The clarity should be improved.
>
>   **Response:** We apologize for the deficiencies in the presentation of some sections of the paper and have made improvements in the updated manuscript. Below, we provide detailed responses to each of the concerns and questions raised:
>
>
> + Clarification of the term "skill".
>
>    We apologize for the unclear expression of the term ``skill" in the initial manuscript. In the updated version, we have clarified that we focus on the task which is composed of several subtasks or subgoals. We define a skill to be a contextual policy that accomplishes a specific part of this overall task. The abstract context serves as a temporally extended signal, representing the current subtask or the next subgoal to reach. This context guides the agent in learning the skill, or in other words, facilitates low-level policy learning, in environments with sparse rewards and long horizons. Once the agent reaches the end of the current skill, it transitions to the next, accompanied by a corresponding shift in context and skill.
>
>   To achieve this, we combine VQ-VAE and VLM to derive the context of a skill, with the VLM providing external semantic information to establish a better context. After training, each state can be mapped to a "skill" context. The context, which can be viewed as a generalized extension of subgoals, helps the agent capture not only the immediate objective but also the broader trajectory context. Our VQ-VAE + VLM framework ensures that task trajectories are segmented in a more meaningful way, providing a proper context for low-level policy learning. We have emphasized these points in the revised manuscript to enhance clarity and precision regarding the term "skill".
>
>
>  + Preliminary writing issues.
>
>     Thank you for pointing out these issues. In the updated version, we have revised the Preliminary section to improve its readability and address the concerns raised.
>
> + Clarity and reference order of Section 5 and Section 6.
>
>   Thank you for your suggestion. In the updated manuscript, we explain that Section 5 theoretically shows how our VLM-guided skill context learning outperforms reinforcement learning in the original space or hierarchical reinforcement learning methods that lack external semantic information guidance. It demonstrates that if the action sequences in the learned skills exhibit tighter temporal correlation, the learning will achieve a smaller suboptimality bound.
>     In the previous version, we introduced Section 6.5 earlier to support these claims by showing how VLM-guided skill context learning reduces the low-level policy search space. To make things clearer, we also adjust the experiment reference order for better coherence in the updated version.
>
> + Equation reference error.
>
>   Thank you for pointing it out. In Line 210, Eq.3 is referred to explain that $\Delta h$ represents the smoothness loss. We have revised the sentence to make it clear.
>
> 2. The illustration is hard to understand.
>
> **Response:** We apologize for the misleading illustration, which may not have fully conveyed our intended idea. The codebook icon in Figure 1 is redundant and does not accurately reflect the low-level policy extraction process. To address this, we have removed the figure in the updated manuscript. Once the skill representations are learned, the high-level policy simply outputs the appropriate skill, and no further reference to the codebook is needed during rollout.
>
> 3. VLM is used during training or rollout?
>
> **Response:** The VLM is used only during training, where it assists in discovering the proper skill context for each state. Specifically, an encoder maps the states to their corresponding skills, enabling the low-level policy to be learned. During the rollout or evaluation phase, the VLM is no longer required, which significantly reduces deployment time while retaining the benefits of its guidance during training. In the ablation study, we analyze the impact of removing the VLM during the training phase.
>
> 4. Explain what is "proper initialization"?
>
> **Response:** We apologize for the insufficient explanation of "proper initialization" in the main text. To clarify, "proper initialization" refers to the process of setting initial trajectory segments before querying the VLM. This approach is necessary because the VLM performs better when labeling shorter segments with the skill context. Additionally, it significantly reduces query time costs.

---

> ### Author Response · Authors · 2024-11-22
>
> 5. Ablations regarding the VAE decoder are missing.
>
> **Response:** We apologize for not clarifying the structure of the decoder. We would like to clarify that the VAE policy decoder is a basic MLP network architecture designed to map the state and skill context to the action. Both the encoder and decoder work together as part of the VQ-VAE loss, with the encoder mapping observations to latent embeddings and the decoder projecting these embeddings into the action space. Therefore, we did not intend to ablate the basic network architecture. While more sophisticated model architectures may offer benefits, they are probably beyond the scope of our work.
>
> 6. How many skills were identified with VLM guidance compared to those without VLM guidance?
>
> **Response:**
> Thank you for raising this question. We now provide a clear comparison of the number of skills identified with and without VLM guidance. Using the inner-built subtask definition in the task as the ground truth, we counted the skills identified across 10 randomly sampled trajectories, and the average value is presented below. These results demonstrate an improvement in skill identification when VLM guidance is applied.
>
> | **Task Name**                           | **VanTA w/o VLM** | **VanTA** |
> |-----------------------------------------|-------------------|-----------|
> | kitchen-complete-v0                     | 1.9               | 2.7       |
> | kitchen-partial-v0                      | 2.6               | 2.8       |
> | kitchen-mixed-v0                        | 2.7               | 3.0       |
> | MiniGrid-DoorKey-6x6-complete-v0        | 2.0               | 2.5       |
> | MiniGrid-DoorKey-6x6-mixed-v0           | 1.7               | 2.6       |
> | MiniGrid-DoorKey-8x8-complete-v0        | 1.4               | 2.5       |
> | MiniGrid-DoorKey-8x8-mixed-v0           | 0.8               | 2.3       |
> | MiniGrid-KeyCorridorS3R3-complete-v0    | 1.4               | 3.1       |
> | MiniGrid-KeyCorridorS3R3-mixed-v0       | 1.2               | 2.7       |
> | MiniGrid-RedBlueDoors-6x6-complete-v0   | 0.7               | 1.7       |
> | MiniGrid-RedBlueDoors-6x6-mixed-v0      | 0.4               | 1.8       |
> | Crafter-partial                         | 0.9               | 3.9       |
>
> 7. Do you have any experiments in a purely IL setting? Or could you provide further motivation for the policy extraction via Q-Learning dependent on a reward function?
>
> **Response:** Since our setup is based on imitation learning, we conducted experiments using datasets that are relatively close to optimal. Specifically, we used the Franka Kitchen environment and expert demonstration data from MiniGrid. In certain environments, the low-level imitation policy performs comparably to the low-level policy trained through offline reinforcement learning.
> The results are presented as follows.
>
> | **Task Name**                         | **IL-VanTA**         | **VanTA**          |
> |---------------------------------------|----------------------|--------------------|
> | kitchen-complete-v0                   | 64.5 ± 4.7           | **69.2 ± 8.5**     |
> | kitchen-partial-v0                    | 48.2 ± 6.6           | **71.2 ± 5.7**     |
> | kitchen-mixed-v0                      | 58.1 ± 3.4           | **68.5 ± 4.4**     |
> | MiniGrid-DoorKey-6x6-complete-v0      | 0.91 ± 0.03          | **0.92 ± 0.03**    |
> | MiniGrid-DoorKey-8x8-complete-v0      | **0.95 ± 0.04**      | 0.92 ± 0.07        |
> | MiniGrid-KeyCorridorS3R3-complete-v0  | **0.84 ± 0.03**      | 0.81 ± 0.06        |
> | MiniGrid-RedBlueDoors-6x6-complete-v0 | 0.78 ± 0.07          | **0.85 ± 0.11**    |
>
> Besides, the motivation for using a Q-learning method to optimize the low-level policy is that our skill context serves as the signal representing the current subtask or the next subgoal to reach. In other words, the skill context acts as a meaningful guide for low-level policy learning, particularly in tasks with sparse rewards or long horizons. Therefore, unlike fixed skills, we aim to dynamically optimize the skill using reward signals from offline data, as long as it benefits offline learning. Our main contribution lies in improving the context of skill through our VLM-guided VQ framework.

---

> ### Author Response · Authors · 2024-11-22
>
> 8. Could you provide some graphical visualization of the table in Section 6.4?
>
> **Response:** Thank you for your suggestion. In response, we have included a graphical visualization in the revised manuscript that demonstrates the slower degradation of VanTA compared to other methods, such as CQL and GCSL. We believe this makes the comparison more apparent and enhances the clarity of our claims, and our graphical visualization is shown in [https://imgur.com/a/tK4x2Y3](https://imgur.com/a/tK4x2Y3).
>
> 9. Does the VQ get a single frame or multiple frames to segment? How does the segmentation work? How to determine the initial and terminal states of a primitive skill?
>
> **Response:** In our work, VQ-VAE is used to encode each frame into a latent embedding within a discrete space, which is the context for skill. As consecutive frames are assigned to the same context in the latent space, the trajectory is segmented into sub-trajectories. The initial and terminal states of a primitive skill are defined as the first and last frames of each sub-trajectory, respectively, as multiple frames share the same skill context. This segmentation process is essential for identifying different skills and lays the foundation for the intervention of external VLM semantic information. We have refined the writing of the segmentation section in the updated version.
>
> We hope these clarifications and revisions address the reviewer’s concerns and improve the overall quality of the paper. Thank you again for your valuable suggestions.

---

### Official Review · Reviewer_rUvx · 2024-11-04

**Soundness:** 3
**Presentation:** 3
**Contribution:** 3
**Rating:** 6
**Confidence:** 5

**Summary:**

The paper proposes using vision-language models (VLMs) to annotate temporally extended skills from offline datasets. Specifically, the annotation is on the latent space after vector quantization and is improved iteratively. It saves the extensive manual labeling process. The proposed method shows robust improvements over the existing state-of-the-art techniques and the authors conduct ablations on the effectiveness of VLM guidance.

**Strengths:**

+ The proposed method utilizes the knowledge of VLMs as accurate skill annotations to iteratively identify helpful low-level skills for various tasks;

+ The annotation from VLMs is done in discrete latent space from the codebook which improves both the learning of the codebook, and the representation of each latent;

+ The paper provides theoretical analysis to guide the algorithm designs;

+ The empirical experiments show decent and robust improvements over existing methods.

**Weaknesses:**

Is it possible to finetune the VLMs to achieve even higher performance since the tasks are quite different from what VLMs are typically trained on (Internet data)? And is it cost-effective to fine-tune VLMs just to label the skills?

**Questions:**

Is it possible to finetune the VLMs to achieve even higher performance since the tasks are quite different from what VLMs are typically trained on (Internet data)? And is it cost-effective to fine-tune VLMs just to label the skills?

---

> ### Author Response · Authors · 2024-11-22
>
> 1. Is it possible to Fine-tune VLM for better performance?
>
> **Response:**
> Thank you for your valuable suggestion. In our work, we adopt GPT-4o as the Vision-Language Model (VLM) because it excels in segmenting task trajectories and identifying semantically meaningful skills. However, GPT-4o is not open source, which limits our ability to fine-tune it directly. To explore alternatives, we consider open-source models such as Llava-Visionary-70B, but even using parameter-efficient fine-tuning methods like LoRA [1], it requires significant computational resources (e.g., 4 H100 GPUs), which exceed the resources available to us.
>
> We also evaluate smaller, more lightweight models such as MobileVLM-3B, which are easier to fine-tune. Unfortunately, when testing this model, we find that it struggles to follow instructions and fails to accurately recognize skills in our tested environments. Fine-tuning such models requires a large amount of high-quality, manually aligned data, which contradicts one of the core objectives of our work: minimizing human labor in the skill extraction process.
>
> We completely agree that fine-tuning enhances performance. With task-specific adaptations, fine-tuned VLMs likely excel in skill recognition. However, even in its pretrained form, GPT-4o demonstrates strong capabilities. With exposure to a wide variety of image-language scenarios during pretraining, it effectively identifies skills in complex tasks, as demonstrated in Section 6.2. This ability makes it a practical and cost-effective choice for our approach.
>
> [1] Hu E J, Shen Y, Wallis P, et al. Lora: Low-rank adaptation of large language models.

---

### Meta-Review · Area_Chair_nBf5 · 2024-12-18

**Metareview:**

This paper presents VanTA, a method for extracting temporally extended skills in reinforcement learning using vision-language models. The paper received mixed initial reviews but ultimately gained stronger support through the discussion phase, with all three reviewers rating it marginally above the acceptance threshold. The reviewers consistently praised the paper's empirical effectiveness across multiple environments, the labor-free skill annotation approach, and the innovative use of VLMs for semantic skill extraction.

Initial concerns focused on three main areas: the novelty of the contribution relative to existing work, unclear presentation of key concepts and implementation details, and questions about the method's generalization capabilities. The authors provided detailed responses addressing these concerns, including additional experimental comparisons with relevant baselines like Choreographer, clarification of the skill definition and VLM integration approach, and new results demonstrating generalization capabilities in the Franka Kitchen environment.

The rebuttal and subsequent discussion were particularly effective in clarifying the technical novelty of VanTA's approach to grounding VLM knowledge in unsupervised skill extraction. The authors convinced reviewers that effectively integrating VLM outputs into the skill extraction loss while maintaining proper learning dynamics represents a non-trivial technical contribution. This was evidenced by improved performance compared to both traditional skill learning methods and approaches requiring human annotation.

**Additional Comments On Reviewer Discussion:**

None -- see metareview

---

### Decision · Program_Chairs · 2025-01-22

Accept (Poster)